# When Does Contrastive Learning Preserve Adversarial Robustness from Pretraining to Finetuning?

**Lijie Fan**[1], **Sijia Liu**[2,3], **Pin-Yu Chen**[3], **Gaoyuan Zhang**[3], **Chuang Gan**[3]
[1] Massachusetts Institute of Technology, [2] Michigan State University,
[3] MIT-IBM Watson AI Lab, IBM Research
`lijiefan@mit.edu, liusiji5@msu.edu,`
{`pin-yu.chen,gaoyuan.zhang,chuangg`}`@ibm.com`

## Abstract

Contrastive learning (CL) can learn generalizable feature representations and achieve state-of-the-art performance of downstream tasks by finetuning a *linear* classifier on top of it. However, as adversarial robustness becomes vital in image classification, it remains unclear whether or not CL is able to preserve robustness to downstream tasks. The main challenge is that in the 'self-supervised pretraining + supervised finetuning' paradigm, adversarial robustness is easily forgotten due to a learning task mismatch from pretraining to finetuning. We call such challenge 'cross-task robustness transferability'. To address the above problem, in this paper we revisit and advance CL principles through the lens of robustness enhancement. We show that (1) the design of contrastive views matters: High-frequency components of images are beneficial to improving model robustness; (2) Augmenting CL with pseudo-supervision stimulus (e.g., resorting to feature clustering) helps preserve robustness without forgetting. Equipped with our new designs, we propose ADVCL, a novel adversarial contrastive pretraining framework. We show that ADVCL is able to enhance cross-task robustness transferability without loss of model accuracy and finetuning efficiency. With a thorough experimental study, we demonstrate that ADVCL outperforms the state-of-the-art self-supervised robust learning methods across multiple datasets (CIFAR-10, CIFAR-100 and STL-10) and finetuning schemes (linear evaluation and full model finetuning). Code is available at `https://github.com/LijieFan/AdvCL`.

## 1 Introduction

Image classification has been revolutionized by convolutional neural networks (CNNs). In spite of CNNs' generalization power, the lack of *adversarial robustness* has shown to be a main weakness that gives rise to security concerns in high-stakes applications when CNNs are applied, e.g., face recognition, medical image classification, surveillance, and autonomous driving [1–5]. The brittleness of CNNs can be easily manifested by generating tiny input perturbations to completely alter the models' decision. Such input perturbations and corresponding perturbed inputs are referred to as *adversarial perturbations* and *adversarial examples (or attacks)*, respectively [6–10].

One of the most powerful defensive schemes against adversarial attacks is adversarial training (AT) [11], built upon a two-player game in which an 'attacker' crafts input perturbations to maximize the training objective for worst-case robustness, and a 'defender' minimizes the maximum loss for an improved robust model against these attacks. However, AT and its many variants using min-max optimization [12–21] were restricted to supervised learning as true labels of training data are required

35th Conference on Neural Information Processing Systems (NeurIPS 2021).

for both supervised classifier and attack generator (that ensures misclassification). The recent work [22–24] demonstrated that with a properly-designed attacker's objective, AT-type defenses can be generalized to the semi-supervised setting, and showed that the incorporation of additional unlabeled data could further improve adversarial robustness in image classification. Such an extension from supervised to semi-supervised defenses further inspires us to ask whether there exist *unsupervised defenses* that can eliminate the prerequisite of labeled data but improve model robustness.

Some very recent literature [25–29] started tackling the problem of adversarial defense through the lens of self-supervised learning. Examples include augmenting a supervised task with an unsupervised 'pretext' task for which ground-truth label is available for 'free' [25, 26], or robustifying unsupervised representation learning based only on a pretext task and then finetuning the learned representations over downstream supervised tasks [27–29]. The latter scenario is of primary interest to us as a defense can then be performed at the pretraining stage without needing any label information. Meanwhile, self-supervised contrastive learning (CL) has been outstandingly successful in the field of representation learning: It can surpass a supervised learning counterpart on downstream image classification tasks in standard accuracy [30–34]. Different from conventional self-supervised learning methods [35], CL, e.g., SimCLR [30], enforces instance discrimination by exploring multiple views of the same data and treating every instance under a specific view as a class of its own [36].

The most relevant work to ours is [27, 28], which integrated adversarial training with CL. However, the achieved adversarial robustness at downstream tasks largely relies on the use of advanced finetuning techniques, either adversarial full finetuning [27] or adversarial linear finetuning [28]. Different from [27, 28], we ask:

*(Q) How to accomplish robustness enhancement using CL without losing its finetuning efficiency, e.g., via a standard linear finetuner?*

Our work attempts to make a rigorous and comprehensive study on addressing the above question. We find that self-supervised learning (including the state-of-the-art CL) suffers a new robustness challenge that we call 'cross-task robustness transferability', which was largely overlooked in the previous work. That is, there exists a task mismatch from pretraining to finetuning (e.g., from CL to supervised classification) so that adversarial robustness is not able to transfer across tasks even if pretraining datasets and finetuning datasets are drawn from the same distribution. Different from supervised/semi-supervised learning, this is a characteristic behavior of self-supervision when being adapted to robust

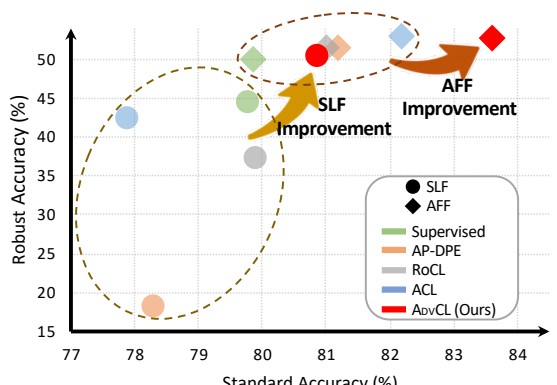

Figure 1: Summary of performance for various robust pretraining methods on CIFAR-10. The covered baseline methods include AP-DPE [26], RoCL [28], ACL [27] and supervised adversarial training (AT) [11]. Upper-right indicates better performance with respect to (w.r.t.) standard accuracy and robust accuracy (under PGD attack with 20 steps and $8/255$ $\ell_\infty$-norm perturbation strength). Different colors represent different pretraining methods, and different shapes represent different finetuning settings. Circles (●) indicates *Standard Linear Finetuning* (SLF), and Diamonds (◆) indicates *Adversarial Full Finetuning* (AFF). Our method (ADVCL, red circle/diamond) has the best performance across finetuning settings. Similar improvement could be observed under Auto-Attacks, and we provide the visualization in the appendix.

learning. As shown in Figure 1, our work advances CL in the adversarial context and the proposed method outperforms all state-of-the-art baseline methods, leading to a substantial improvement in both robust accuracy and standard accuracy using either the lightweight standard linear finetuning or end-to-end adversarial full finetuning.

**Contributions**   Our main contributions are summarized below.

❶ We propose ADVCL, a unified adversarial CL framework, and propose to use original adversarial examples and high-frequency data components to create robustness-aware and generalization-aware views of unlabeled data.

❷ We propose to generate proper pseudo-supervision stimulus for ADVCL to improve cross-task robustness transferability. Different from existing self-supervised defenses aided with labeled data [27], we generate pseudo-labels of unlabeled data based on their clustering information.

❸ We conduct a thorough experimental study and show that ADVCL achieves state-of-the-art robust accuracies under both PGD attacks [11] and Auto-Attacks [37] using *only standard linear finetuning*. For example, in the case of Auto-Attack (the most powerful threat model) with $8/255\ \ell_\infty$-norm perturbation strength under ResNet-18, we achieve $3.44\%$ and $3.45\%$ robustness improvement on CIFAR-10 and CIFAR-100 over existing self-supervised methods. We also justify the effectiveness of ADVCL in different attack setups, dataset transferring, model explanation, and loss landscape smoothness.

## 2  Background & Related Work

**Self-Supervised Learning** Early approaches for unsupervised representation learning leverages handcrafted tasks, like prediction rotation [38] and solving the Jigsaw puzzle [39, 40], geometry prediction [41] and Selfie [42]. Recently contrastive learning (CL) [30, 33, 34, 43–45] and its variants [31, 32, 36, 46] have demonstrated superior abilities in learning generalizable features in an unsupervised manner. The main idea behind CL is to self-create positive samples of the same image from aggressive viewpoints, and then acquire data representations by maximizing agreement between positives while contrasts with negatives.

In what follows, we elaborate on the **formulation of SimCLR** [30], one of the most commonly-used CL frameworks, which this paper will focus on. To be concrete, let $\mathcal{X} = \{x_1, x_2, ..., x_n\}$ denote an *unlabeled source* dataset, SimCLR offers a learned *feature encoder* $f_\theta$ to generate expressive deep representations of the data. To train $f_\theta$, each input $x \in \mathcal{X}$ will be transformed into two *views* $(\tau_1(x), \tau_2(x))$ and labels them as a positive pair. Here transformation operations $\tau_1$ and $\tau_2$ are randomly sampled from a pre-defined transformation set $\mathcal{T}$, which includes, e.g., random cropping and resizing, color jittering, rotation, and cutout. The positive pair is then fed in the feature encoder $f_\theta$ with a projection head $g$ to acquire projected features, i.e., $z_i = g \circ f_\theta(\tau_i(x))$ for $j \in \{1, 2\}$. *NT-Xent loss* (i.e., the normalized temperature-scaled cross-entropy loss) is then applied to optimizing $f_\theta$, where the distance of projected positive features $(z_1, z_2)$ is minimized for each input $x$. SimCLR follows the '*self-supervised pretraining + supervised finetuning*' paradigm. That is, once $f_\theta$ is trained, a downstream supervised classification task can be handled by just finetuning a linear classifier $\phi$ over the fixed encoder $f_\theta$, leading to the eventual classification network $\phi \circ f_\theta$.

**Adversarial Training (AT)** Deep neural networks are vulnerable to adversarial attacks. Various approaches have been proposed to enhance the model robustness. Given a classification model $\theta$, AT [11] is one of the most powerful robust training methods against adversarial attacks. Different from standard training over normal data $(x, y) \in \mathcal{D}$ (with feature $x$ and label $y$ in dataset $\mathcal{D}$), AT adopts a min-max training recipe, where the worst-case training loss is minimized over the adversarially perturbed data $(x + \delta, y)$. Here $\delta$ denotes the input perturbation variable to be maximized for the worst-case training objective. The *supervised AT* is then formally given by

$$\min_\theta \mathbb{E}_{(x,y)\in D} \max_{\|\delta\|_\infty \leq \epsilon} \ell(x + \delta, y; \theta), \tag{1}$$

where $\ell$ denotes the supervised training objective, e.g., cross-entropy (CE) loss. There have been many variants of AT [19–21, 47–50, 22–25] established for supervised/semi-supervised learning.

**Self-supervision enabled AT** Several recent works [26–29] started to study how to improve model robustness using *self-supervised AT*. Their idea is to apply AT (1) to a self-supervised pretraining task, e.g., SimCLR in [27, 28], such that the learned feature encoder $f_\theta$ renders robust data representations. However, different from our work, the existing ones lack a systematic study on *when* and *how* self-supervised robust pretraining can preserve robustness to downstream tasks without sacrificing the efficiency of lightweight finetuning. For example, the prior work [26, 27] suggested adversarial full finetuning, where pretrained model is used as a weight initialization in finetuning downstream tasks. Yet, it requests the finetuner to update all of the weights of the pretrained model, and thus makes the advantage of self-supervised robust pretraining less significant. A more practical scenario is *linear finetuning*: One freezes the pretrained feature encoder for the downstream task and only partially finetunes a linear prediction head. The work [28] evaluated the performance of linear fintuning but observed a relatively large performance gap between the *standard* linear finetuning and *adversarial* linear finetuning; see more comparisons in Figure 1. Therefore, the problem–*how to enhance robustness transferability from pretraining to linear finetuning*–remains unexplored.

# 3 Problem Statement

In this section, we present the problem of our interest, together with its setup.

**Robust pretraining + linear finetuning.** We aim to develop robustness enhancement solutions by fully exploiting and exploring the power of CL at the pretraining phase, so that the resulting robust feature representations can seamlessly be used to generate robust predictions of downstream tasks using just a lightweight finetuning scheme. With the aid of AT (1), we formulate the '*robust pretraining + linear finetuning*' problem below:

$$\text{Pretraining:} \min_{\theta} \mathbb{E}_{x \in \mathcal{X}} \max_{\|\delta\|_{\infty} \leq \epsilon} \ell_{\text{pre}}(x + \delta, x; \theta) \tag{2}$$

$$\text{Finetuning:} \min_{\theta_c} \mathbb{E}_{(x,y) \in \mathcal{D}} \ell_{\text{CE}}(\phi_{\theta_c} \circ f_{\theta}(x), y), \tag{3}$$

where $\ell_{\text{pre}}$ denotes a properly-designed robustness- and generalization-aware CL loss (see Sec. 4) given as a function of the adversarial example $(x + \delta)$, original example $x$ and feature encoder parameters $\theta$. In (2), $\phi_{\theta_c} \circ f_{\theta}$ denotes the classifier by equipping the linear prediction head $\phi_{\theta_c}$ (with parameters $\theta_c$ to be designed) on top of the fixed feature encoder $f_{\theta}$, and $\ell_{\text{CE}}$ denotes the supervised CE loss over the target dataset $\mathcal{D}$. Note that besides the standard linear finetuning (3), one can also modify (3) using the worst-case CE loss for adversarial linear/full finetuning [27, 28]. We do not consider standard full finetuning in the paper since tuning the full network weights with standard cross-entropy loss is not possible for the model to preserve robustness [26].

**Cross-task robustness transferability.** Different from supervised/semi-supervised learning, self-supervision enables robust pretraining over *unlabeled* source data. In the meantime, it also imposes a new challenge that we call '*cross-task robustness transferability*': At the pretraining stage, a feature encoder is learned over a 'pretext' task for which ground-truth is available for free, while finetuning is typically carried out on a new downstream task. Spurred by the above, we ask the following questions:

- Will CL improve adversarial robustness using just standard linear finetuning?
- What are the principles that CL should follow to preserve robustness across tasks?
- What are the insights can we acquire from self-supervised robust representation learning?

# 4 Proposed Approach: Adversarial Contrastive Learning (ADVCL)

In this section, we develop a new adversarial CL framework, ADVCL, which includes two main components, robustness-aware view selection and pseudo-supervision stimulus generation. In particular, we advance the view selection mechanism by taking into account proper frequency-based data transformations that are beneficial to robust representation learning and pretraining generalization ability. Furthermore, we propose to design and integrate proper supervision stimulus into ADVCL so as to improve the cross-task robustness transferability since robust representations learned from self-supervision may lack the class-discriminative ability required for robust predictions on downstream tasks. We provide an overview of ADVCL in Figure 2.

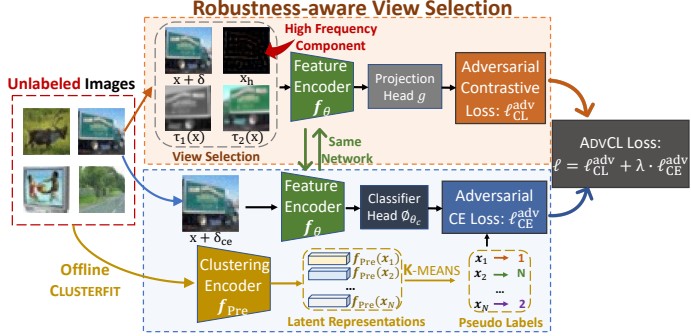

Figure 2: The overall pipeline of ADVCL. It mainly has two ingredients: robustness-aware view selection (orange box) and pseudo-supervision stimulus generation (blue box). The view selection mechanism is advanced by high frequency components, and the supervision stimulus is created by generating pseudo labels for each image through CLUSTERFIT. The pseudo label (in yellow color) can be created in an offline manner and will not increase the computation overhead.

## 4.1 View selection mechanism

In contrast to standard CL, we propose two additional contrastive views: the adversarial view and the frequency view, respectively.

**Multi-view CL loss**   Prior to defining new views, we first review the NT-Xent loss and its multi-view version used in CL. Following notations defined in Sec. 2, the contrastive loss with respect to (w.r.t.) a positive pair $(\tau_1(x), \tau_2(x))$ of each (unlabeled) data $x$ is given by

$$\ell_{\mathrm{CL}}(\tau_1(x), \tau_2(x)) = -\sum_{i=1}^{2} \sum_{j \in \mathcal{P}(i)} \log \frac{\exp\big(\mathrm{sim}(z_i, z_j)/t\big)}{\sum_{k \in \mathcal{N}(i)} \exp\big(\mathrm{sim}(z_i, z_k)/t\big)}, \qquad (4)$$

where recall that $z_i = g \circ f(\tau_i(x))$ is the projected feature under the $i$th view, $\mathcal{P}(i)$ is the set of positive views except $i$ (e.g., $\mathcal{P}(i) = \{2\}$ if $i = 1$), $\mathcal{N}(i)$ denotes the set of augmented batch data except the point $\tau_i(x)$, the cardinality of $\mathcal{N}(i)$ is $(2b - 1)$ (for a data batch of size $b$ under 2 views), $\mathrm{sim}(z_{i1}, z_{i2})$ denotes the cosine similarity between representations from two views of the same data, $\exp$ denotes exponential function, $\mathrm{sim}(\cdot, \cdot)$ is the cosine similarity between two points, and $t > 0$ is a temperature parameter. The two-view CL objective can be further extend to the *multi-view contrastive loss* [51]

$$\ell_{\mathrm{CL}}(\tau_1(x), \tau_2(x), \ldots, \tau_m(x)) = -\sum_{i=1}^{m} \sum_{j \in \mathcal{P}(i)} \log \frac{\exp\big(\mathrm{sim}(z_i, z_j)/t\big)}{\sum_{k \in \mathcal{N}(i)} \exp\big(\mathrm{sim}(z_i, z_k)/t\big)}, \qquad (5)$$

where $\mathcal{P}(i) = [m]/\{i\}$ denotes the $m$ positive views except $i$, $[m]$ denotes the integer set $\{1, 2, \ldots, m\}$, and $\mathcal{N}(i)$, with cardinality $(bm - 1)$, denotes the set of $m$-view augmented $b$ batch samples except the point $\tau_i(x)$.

**Contrastive view from adversarial example**   Existing methods proposed in [27–29] can be explained based on (4): An adversarial perturbation $\boldsymbol{\delta}$ w.r.t. each view of a sample $x$ is generated by maximizing the contrastive loss:

$$\delta_1^*, \delta_2^* = \underset{\|\delta_i\|_\infty \leq \epsilon}{\operatorname{argmax}} \, \ell_{\mathrm{CL}}(\tau_1(x) + \delta_1, \tau_2(x) + \delta_2). \qquad (6)$$

A solution to problem (6) eventually yields a *paired* perturbation view $(\tau_1(x) + \delta_1^*, \tau_2(x) + \delta_2^*)$. However, the definition of adversarial view (6) used in [27–29] may not be proper. First, standard CL commonly uses *aggressive* data transformation that treats small portions of images as positive samples of the full image [36]. Despite its benefit to promoting generalization, crafting perturbations over such aggressive data transformations may not be suitable for defending adversarial attacks applied to *full* images in the adversarial context. Thus, a new adversarial view built upon $x$ rather than $\tau_i(x)$ is desired. Second, the contrastive loss (4) is only restricted to two views of the same data. As will be evident later, the multi-view contrastive loss is also needed when taking into account multiple robustness-promoting views. Spurred by above, we define the *adversarial view* over $x$, without modifying the existing data augmentations $(\tau_1(x), \tau_2(x))$. This leads to the following adversarial perturbation generator by maximizing a 3-view contrastive loss

$$\delta^* = \underset{\|\delta\| \leq \epsilon}{\operatorname{argmax}} \, \ell_{\mathrm{CL}}(\tau_1(x), \tau_2(x), x + \delta), \qquad (7)$$

where $x + \delta^*$ is regarded as the third view of $x$.

**Contrastive view from high-frequency component**   Next, we use the high-frequency component (HFC) of data as another additional contrastive view. The rationale arises from the facts that 1) learning over HFC of data is a main cause of achieving superior generalization ability [52] and 2) an adversary typically concentrates on HFC when manipulating an example to fool model's decision [53]. Let $\mathcal{F}$ and $\mathcal{F}^{-1}$ denote Fourier transformation and its inverse. An input image $x$ can then be decomposed into its HFC $x_{\mathrm{h}}$ and low-frequency component (LFC) $x_{\mathrm{l}}$:

$$x_{\mathrm{h}} = \mathcal{F}^{-1}(q_{\mathrm{h}}), \quad x_{\mathrm{l}} = \mathcal{F}^{-1}(q_{\mathrm{l}}), \quad [q_{\mathrm{h}}, q_{\mathrm{l}}] = \mathcal{F}(x). \qquad (8)$$

In (8), the distinction between $q_{\mathrm{h}}$ and $q_{\mathrm{l}}$ is made by a hard thresholding operation. Let $q(i, j)$ denote the $(i, j)$th element of $\mathcal{F}(x)$, and $c = (c_1, c_2)$ denote the centriod of the frequency spectrum. The components $q_{\mathrm{l}}$ and $q_{\mathrm{h}}$ in (8) are then generated by filtering out values according to the distance from $c$: $q_h(i, j) = \mathbb{1}_{[d((i,j),(c_1,c_2)) \geq r]} \cdot q(i, j)$, and $q_l(i, j) = \mathbb{1}_{[d((i,j),(c_1,c_2)) \leq r]} \cdot q(i, j)$, where $d(\cdot, \cdot)$ is the Euclidian distance between two spatial coordinates, $r$ is a pre-defined distance threshold ($r = 8$ in all our experiments), and $\mathbb{1}_{[\cdot]} \in \{0, 1\}$ is an indicator function which equals to 1 if the condition within $[\cdot]$ is met and 0 otherwise.

**Robustness-aware contrastive learning objective**  By incorporating the adversarial perturbation $\delta$ and disentangling HFC $x_\mathrm{h}$ from the original data $x$, we obtain a four-view contrastive loss (5) defined over $(\tau_1(x), \tau_2(x), x + \delta, x_\mathrm{h})$,

$$\ell_{\mathrm{CL}}^{\mathrm{adv}}(\theta; \mathcal{X}) := \mathbb{E}_{x \in \mathcal{X}} \max_{\|\delta\|_\infty \leq \epsilon} \ell_{\mathrm{CL}}(\tau_1(x), \tau_2(x), x + \delta, x_\mathrm{h}; \theta), \tag{9}$$

where recall that $\mathcal{X}$ denotes the unlabeled dataset, $\epsilon > 0$ is a perturbation tolerance during training, and for clarity, the four-view contrastive loss (5) is explicitly expressed as a function of model parameters $\theta$. As will be evident latter, the eventual learning objective ADVCL will be built upon (9).

### 4.2  Supervision stimulus generation: ADVCL empowered by CLUSTERFIT

On top of (9), we further improve the robustness transferability of learned representations by generating a proper supervision stimulus. Our rationale is that robust representation could lack the class-discriminative power required by robust classification as the former is acquired by optimizing an unsupervised contrastive loss while the latter is achieved by a supervised cross-entropy CE loss. However, there is no knowledge about supervised data during pretraining. In order to improve cross-task robustness transferability but without calling for supervision, we take advantage of CLUSTERFIT [54], a pseudo-label generation method used in representation learning.

To be more concrete, let $f_{\mathrm{pre}}$ denote a pretrained representation network that can generate latent features of unlabeled data. Note that $f_{\mathrm{pre}}$ can be set available beforehand and trained over either supervised or unsupervised dataset $\mathcal{D}_{\mathrm{pre}}$, e.g., ImageNet using using CL in experiments. Given (normalized) pretrained data representations $\{f_{\mathrm{pre}}(x)\}_{x \in \mathcal{X}}$, CLUSTERFIT uses $K$-*means clustering* to find $K$ data clusters of $\mathcal{X}$, and maps a *cluster index* $c$ to a *pseudo-label*, resulting in the pseudo-labeled dataset $\{(x, c) \in \hat{\mathcal{X}}\}$. By integrating CLUSTERFIT with (9), the eventual training objective of ADVCL is then formed by

$$\min_\theta \ell_{\mathrm{CL}}^{\mathrm{adv}}(\theta; \mathcal{X}) + \lambda \underbrace{\min_{\theta, \theta_\mathrm{c}} \mathbb{E}_{(x,c) \in \hat{\mathcal{X}}} \max_{\|\delta_{ce}\|_\infty \leq \epsilon} \ell_{\mathrm{CE}}(\phi_{\theta_\mathrm{c}} \circ f_\theta(x + \delta_{ce}), c)}_{\text{Pseudo-classification enabled AT regularization}}, \tag{10}$$

where $\hat{\mathcal{X}}$ denotes the pseudo-labeled dataset of $\mathcal{X}$, $\phi_{\theta_\mathrm{c}}$ denotes a prediction head over $f_\theta$, and $\lambda > 0$ is a regularization parameter that strikes a balance between adversarial contrastive training and pseudo-label stimulated AT. When the number of clusters $K$ is not known *a priori*, we extend (10) to an *ensemble version* over $n$ choices of cluster numbers $\{K_1, \ldots, K_n\}$. Here each cluster number $K_i$ is paired with a unique linear classifier $\phi_i$ to obtain the supervised prediction $\phi_i \circ f$ (using cluster labels). The ensemble CE loss, given by the average of $n$ individual losses, is then used in (10). Our experiments show that the ensemble version usually leads to better generalization ability.

## 5  Experiments

In this section, we demonstrate the effectiveness of our proposed ADVCL from the following aspects: (1) Quantitative results, including cross-task robustness transferability, cross-dataset robustness transferability, and robustness against PGD attacks [11] and Auto-Attacks [37]; (2) Qualitative results, including representation t-SNE [55], feature inversion map visualization, and geometry of loss landscape; (3) Ablation studies of ADVCL, including finetuning schemes, view selection choices, and supervision stimulus variations.

**Experiment setup**  We consider three robustness evaluation metrics: (1) Auto-attack accuracy (**AA**), namely, classification accuracy over adversarially perturbed images via Auto-Attacks; (2) Robust accuracy (**RA**), namely, classification accuracy over adversarially perturbed images via PGD attacks; and (3) Standard accuracy (**SA**), namely, standard classification accuracy over benign images without perturbations. We use ResNet-18 for the encoder architecture of $f_\theta$ in CL. Unless specified otherwise, we use 5-step $\ell_\infty$ projected gradient descent (PGD) with $\epsilon = 8/255$ to generate perturbations during pretraining, and use Auto-Attack and 20-step $\ell_\infty$ PGD with $\epsilon = 8/255$ to generate perturbations in computing AA and RA at test time. We will compare ADVCL with the CL-based adversarial pretraining **baselines** , ACL [27], RoCL [28], (non-CL) self-supervised adversarial learning baseline AP-DPE [26] and the supervised AT baseline [11].

### 5.1  Quantitative Results

**Overall performance from pretraining to finetuning (across tasks)**  In Table 1, we evaluate the robustness of a classifier (ResNet-18) finetuned over robust representations learned by different

Table 1: Cross-task performance of ADVCL (in dark gray color), compared with supervised (in white color) and self-supervised (in light gray color) baselines, in terms of AA, RA and SA on CIFAR-10 with ResNet-18. The pretrained models are evaluated under the standard linear finetuning (SLF) setting and the adversarial full finetuning (AFF) setting. The top performance is highlighted in **bold**.

| Pretraining Method | Finetuning Method | CIFAR-10 | | | CIFAR-100 | | |
|---|---|---|---|---|---|---|---|
| | | AA(%) | RA(%) | SA(%) | AA(%) | RA(%) | SA(%) |
| Supervised | Standard linear finetuning (**SLF**) | 42.22 | 44.4 | 79.77 | 19.53 | 23.41 | **50.53** |
| AP-DPE[26] | | 16.07 | 18.22 | 78.30 | 4.17 | 6.23 | 47.91 |
| RoCL[28] | | 28.38 | 39.54 | 79.90 | 8.66 | 18.79 | 49.53 |
| ACL[27] | | 39.13 | 42.87 | 77.88 | 16.33 | 20.97 | 47.51 |
| **ADVCL (ours)** | | **42.57** | **50.45** | **80.85** | **19.78** | **27.67** | 48.34 |
| Supervised | Adversarial full finetuning (**AFF**) | 46.19 | 49.89 | 79.86 | 21.61 | 25.86 | 52.22 |
| AP-DPE[26] | | 48.13 | 51.52 | 81.19 | 22.53 | 26.89 | 55.27 |
| RoCL[28] | | 47.88 | 51.35 | 81.01 | 22.38 | 27.49 | 55.10 |
| ACL[27] | | 49.27 | **52.82** | 82.19 | 23.63 | **29.38** | 56.61 |
| **ADVCL (ours)** | | **49.77** | 52.77 | **83.62** | **24.72** | 28.73 | **56.77** |

supervised/self-supervised pretraining approaches over CIFAR-10 and CIFAR-100. We focus on two representative finetuning schemes: the simplest standard linear finetuning (SLF) and the end-to-end adversarial full finetuning (AFF). As we can see, the proposed ADVCL method yields a substantial improvement over almost all baseline methods. Moreover, ADVCL improves robustness and standard accuracy simultaneously.

**Robustness transferability across datasets** In Table 2, we next evaluate the robustness transferability across different datasets, where $A \rightarrow B$ denotes the transferability from pretraining on dataset $A$ to finetuning on another dataset $B (\neq A)$ of representations learned by ADVCL. Here the pretraining setup is consistent with Table 1. We observe that ADVCL yields

Table 2: Cross-dataset performance of ADVCL (dark gray color), compared with supervised (white color) and self-supervised (light gray) baselines, in AA, RA, SA, on STL-10 with ResNet-18.

| Method | Fine-tuning | CIFAR-10 → STL-10 | | | CIFAR-100 → STL-10 | | |
|---|---|---|---|---|---|---|---|
| | | AA(%) | RA(%) | SA(%) | AA(%) | RA(%) | SA(%) |
| Supervised | **SLF** | 22.26 | 30.45 | 54.70 | 19.54 | 23.63 | **51.11** |
| RoCL[28] | | 18.65 | 28.18 | 54.56 | 12.39 | 21.93 | 47.86 |
| ACL[27] | | 25.29 | 31.80 | 55.81 | **21.75** | 26.32 | 45.91 |
| **ADVCL (ours)** | | **25.74** | **35.80** | **63.73** | 20.86 | **30.35** | 50.71 |
| Supervised | **AFF** | 33.10 | 36.7 | 62.78 | 29.18 | 32.43 | 55.85 |
| RoCL[28] | | 29.40 | 34.65 | 61.75 | 27.55 | 31.38 | 57.83 |
| ACL[27] | | 32.50 | 35.93 | 62.65 | 28.68 | 32.41 | 57.16 |
| **ADVCL (ours)** | | **34.70** | **37.78** | **63.52** | **30.51** | **33.70** | **61.56** |

better robustness as well as standard accuracy than almost all baseline approaches under both SLF and AFF finetuning settings. In the case of CIFAR-100 → STL-10, although ADVCL yields $0.89\%$ AA drop compared to ACL [27], it yields a much better SA with $4.8\%$ improvement.

**Robustness evaluation vs. attack strength** It was shown in [56] that an adversarial defense that causes *obfuscated* gradients results in a *false sense of model robustness*. The issue of obfuscated gradients typically comes with two 'side effects': (a) The success rate of PGD attack ceases to be improved as the $\ell_\infty$-norm perturbation radius $\epsilon$ increases; (b) A larger number of PGD steps fails to generate stronger adversarial examples. Spurred by the above, Figure 3 shows the finetuning performance of ADVCL (using SLF) as a function of the perturbation size $\epsilon$ and the PGD step number. As we can see, ADVCL is consistently more robust than the baselines at all different PGD settings for a significant margin.

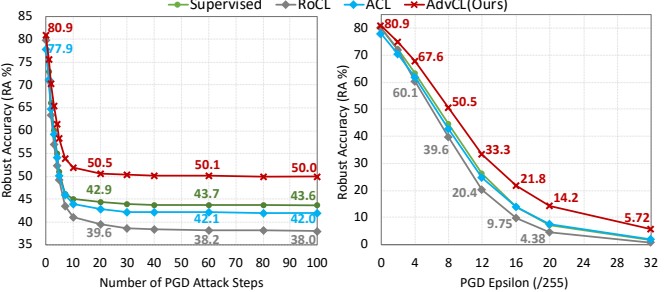

Figure 3: RA of ADVCL and baseline approaches under various PGD attacks. SLF is applied to the pretrained model.

## 5.2 Qualitative Results

**Class discrimination of learned representations**    To further demonstrate the efficacy of ADVCL, Figure 4 visualizes the representations learned by self-supervision using t-SNE [55] on CIFAR-10. We color each point using its ground-truth label. The results show representations learned by ADVCL have a much clearer class boundary than those learned with baselines. This indicates that ADVCL makes an adversary difficult to successfully perturb an image, leading to a more robust prediction.

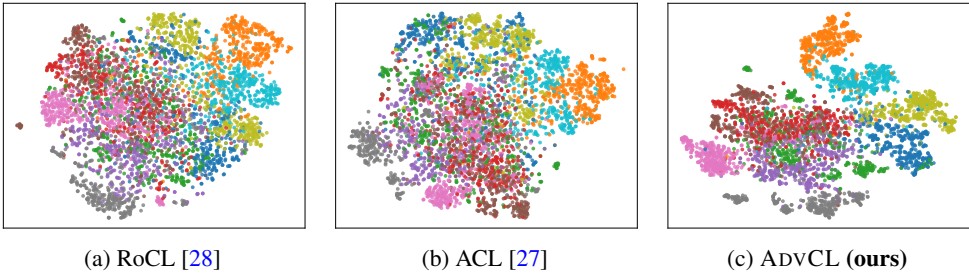

(a) RoCL [28]          (b) ACL [27]          (c) ADVCL **(ours)**

Figure 4: t-SNE visualization of representations learned with different self-supervised pretraining approaches. Our ADVCL gives a much clearer separation among classes than baseline approaches.

**Visual interpretability of learned representations**    Furthermore, we demonstrate the advantage of our proposals from the perspective of model explanation, characterized by feature inversion map (FIM) [57] of internal neurons' response. The work [18, 58, 59] showed that model robustness offered by supervised AT and its variants enforces hidden neurons to learn perceptually-aligned data features through the lens of FIM. However, it remains unclear whether or not *self-supervised* robust pretraining is able to render explainable internal response. Following [57, 58], we acquire FIM of the $i$th component of representation vector by solving the optimization problem $x_{\text{FIM}} = \min_\Delta [f_\theta(x_0 + \Delta)]_i$, where $x_0$ is a randomly selected seed image, and $[\cdot]_i$ denotes the $i$th coordinate of a vector. Figure 5 shows that

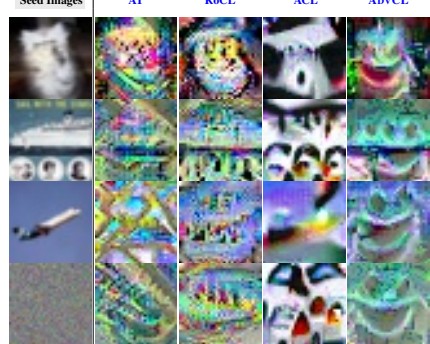

Figure 5: FIM visualization of neuron $502$ under CIFAR-10 using different robust training methods. Column 1 contains different seed images to generate FIM. Columns 2-5 are FIMs using models trained with different approaches.

compared to other approaches, more similar texture-aligned features can be acquired from a neuron's feature representation of the network trained with our method regardless of the choice of seed images.

**Flatter loss landscape implies better transferability**    It has been shown in [60] that the flatness of loss landscape is a good indicator for superb transferability in the pretraining + finetuning paradigm. Motivated by that, Figure 6 presents the adversarial loss landscape of ADVCL and other self-supervised pretraining approaches under SLF, where the loss landscape is drawn using the method in [61]. Note that instead of standard CE loss, we visualize the adversarial loss w.r.t. model weights. As we can see, the loss for ADVCL has a much flatter landscape around the local optima, whereas the losses for the other approaches change more rapidly. This justifies that our proposal has a better robustness transferability than baseline approaches.

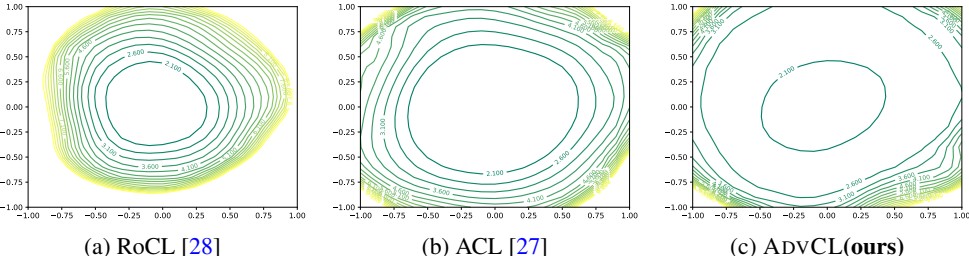

(a) RoCL [28]          (b) ACL [27]          (c) ADVCL**(ours)**

Figure 6: Visualization of adversarial loss landscape w.r.t. model weights using different self-supervised pretraining methods. ADVCL gives a much flatter landscape than the other baselines.

Table 3: Performance (RA and SA) of ADVCL (in dark gray color) and baseline approaches on CIFAR-10, under different linear finetuning strategies: SLF and adversarial linear finetuning (ALF).

| Method | SLF | | ALF | |
|---|---|---|---|---|
| | RA(%) | SA(%) | RA(%) | SA(%) |
| Supervised | 44.40 | 79.77 | 46.75 | 79.06 |
| RoCL[28] | 39.54 | 79.90 | 43.11 | 77.33 |
| ACL[27] | 42.87 | 77.88 | 45.40 | 77.71 |
| **ADVCL(ours)** | **50.45** | **80.85** | **52.01** | **79.39** |

Table 4: Performance (RA and SA) of ADVCL using different contrastive views setups. ResNet-18 is the backbone network, CIFAR-10 is the dataset, and SLF is used for classification.

| Contrastive Views | RA(%) | SA(%) |
|---|---|---|
| $\tau_1(x) + \delta_1, \tau_2(x)$ | 42.12 | 77.07 |
| $\tau_1(x) + \delta_1, \tau_2(x) + \delta_2$ | 42.48 | 73.12 |
| $\tau_1(x) + \delta_1, \tau_2(x) + \delta_2, \tau_1(x), \tau_2(x)$ | 43.51 | 74.22 |
| $x + \delta, \tau_1(x), \tau_2(x)$ | **50.19** | **80.17** |
| $x + \delta, \tau_1(x), \tau_2(x), x_l$ | 49.51 | 79.83 |
| $x + \delta, \tau_1(x), \tau_2(x), x_l, x_h$ | 50.03 | 80.14 |
| $x + \delta, \tau_1(x), \tau_2(x), x_h$ | **50.45** | **80.85** |

Table 5: Performance (RA and SA) of ADVCL using various pretrained models $f_{pre}$ and cluster numbers $K$ in CLUSTERFIT, as well as the baseline w/o using CLUSTERFIT. The setup of $f_{pre}$ is specified by the training method (supervised training or SimCLR) and training dataset (ImageNet or CIFAR-10). ADVCL is implemented using unlabeled data from CIFAR-10 under ResNet-18, together with SLF over the acquired feature encoder for supervised CIFAR-10 classification.

| $f_{pre}$ setup: (dataset, training) | Cluster number $K$ | RA (%) | SA (%) |
|---|---|---|---|
| N/A | w/o CLUSTERFIT | 48.89 | 77.73 |
| (CIFAR-10, SimCLR) | 10 | 50.10 | 80.34 |
| | 100 | 49.21 | 79.52 |
| (ImageNet, supervised) | 10 | 50.16 | 78.27 |
| | 100 | 49.27 | 78.08 |
| (ImageNet, SimCLR) | 2 | 50.09 | 79.72 |
| | 10 | 50.12 | 79.93 |
| | 50 | 49.27 | 79.55 |
| | 100 | 49.16 | 79.07 |
| | 500 | 49.03 | 78.96 |
| | **Ensemble** | **50.45** | **80.85** |

## 5.3 Ablation studies

**Linear finetuning types** We first study the robustness difference when different linear finetuning strategies: *Standard* linear finetuning (SLF) and *Adversarial* linear finetuning (ALF) are applied. Table 3 shows the performance of models trained with different pretraining methods. As we can see, our ADVCL achieves the best performance under both linear finetuning settings and outperforms baseline approaches in a large margin. We also note the performance gap between SLF and ALF induced by our proposal ADVCL is much smaller than other approaches, and ADVCL with SLF achieves much better performance than baseline approaches with ALF. This indicates that the representations learned by ADVCL is already sufficient to yield satisfactory robustness.

**View selection setup** We illustrate how different choices of contrastive views influence the robustness performance of ADVCL in Table 4. The first 4 rows study the effect of different types of adversarial examples in contrastive views, and our proposed 3-view contrastive loss (7) significantly outperforms the other baselines, as shown in row 4. The rows in gray show the performance of further exploring different image frequency components (8) as different contrastive views. It is clear that the use of HFC leads to the best overall performance, as shown in the last row.

**Supervision stimulus setup** We further study the performance of ADVCL using different supervision stimulus. Specifically, we vary the pretrained model for $f_{pre}$ and pseudo cluster number $K$ when training ADVCL and summarize the results in Table 5. The results demonstrate that adding the supervision stimulus could boost the performance of ADVCL. We also observe that the best result comes from $f_{pre}$ pretrained on Imagenet using SimCLR. This is because such representations could generalize better. Moreover, the ensemble scheme over pseudo label categories $K \in \{2, 10, 50, 100, 500\}$ yields better results than using a single number of clusters. The ensemble scheme also makes ADVCL less sensitive to the actual number of labels for the training dataset.

## 6 Conclusion

In this paper, we study the good practices in making contrastive learning robust to adversarial examples. We show that adding perturbations to original images and high-frequency components are two beneficial factors. We further show that proper supervision stimulus could improve model robustness. Our proposed approaches can achieve state-of-the-art robust accuracy as well as standard accuracy using just standard linear finetuning. Extensive experiments involving quantitative and qualitative analysis have also been made not only to demonstrate the effectiveness of our proposals but also to rationalize why it yields superior performance. Future works could be done to improve the scalability of our proposed self-supervised pretraining approach to very large datasets and models to further boost robust transferabilty across datasets.

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
