# Appendices

This supplementary material provides additional implementation details and experimental results.

## A. Discussion and Broader Impact

In this paper we propose a powerful framework, ADVCL, which could preserve robustness from pretraining to finetuning, and we empirically show that the light-weight standard linear finetuning is already sufficient to give us comparable performance to the computational-expensive adversarial full finetuning. We don't think our work would have negative societal impacts. The potential broader impact of our work is that, with the help of our proposed pretraining design paradigm, neural models could preserve adversarial robustness using lightweight linear finetuners, which could be deployed to embodied systems and can make real-time applications more secure and trustworthy on mobile devices.

## B. Implementation Details

**Pretraining Details**    We list implementation details for ADVCL pretraining in this section. We use SGD optimizer with batch size=512, initial learning rate=0.5, momentum=0.9, and weight decay=0.0001 to train the network for 1000 epochs. We use cosine learning rate decay during training, and we use the first 10 epochs to warm-up learning rate from 0.01 to 0.5. The temperature parameter in contrastive loss $\ell_{\mathrm{CL}}$ is set to $t = 0.5$, and the regularization parameter for the cross-entropy term with pseudo labels in Eq.(10) is set to $\lambda = 0.2$. All experiments are performed on 4 NVIDIA TITAN Xp GPUs.

The augmentation set $\mathcal{T}$ for pretraining consists of random cropping with scale 0.2 to 1, random horizontal flip, random color jittering and random grayscale. We provide the pseudo code for implementing $\mathcal{T}$ here in PyTorch in Algorithm A1.

**Finetuning Details**    For SLF, the encoder parameters $f_\theta$ are fixed and the linear classifier is trained for 25 epochs using SGD with batch size=512, initial learning rate=0.1, momentum=0.9, and weight decay=0.0002. The learning rate is decreased to $0.1\times$ at epoch 15, 20. For ALF, we use 10-step $\ell_\infty$ PGD attack with $\epsilon = 8/255$ to generate adversarial perturbations during training. The encoder parameters $f_\theta$ are fixed and the linear classifier is trained for 25 epochs using SGD with batch size=512, initial learning rate=0.1. The learning rate is decreased to $0.1\times$ at epoch 15, 20. For AFF, following the settings in [26], we also use 10-step $\ell_\infty$ PGD attack with $\epsilon = 8/255$ to generate adversarial perturbations during training, and train the entire network parameters $f_\theta$ and the linear classifier with trades loss for 25 epochs with initial learning rate of 0.1 which decreases to $0.1\times$ at epoch 15, 20. We report the AA, RA and SA for the best possible model for every method under every setting.

**TRIBN: Customized batch normalization**    It has recently been shown in [26, 58, 59] that batch normalization (BN) could play a vital role in robust training with 'mixed' normal and adversarial data. Thus, a careful study on the BN strategy of ADVCL is needed, since *two types of adversarial perturbations* are generated in Eq.(10) w.r.t. different adversary's goals, maximizing the CL loss ($\delta$) vs. maximizing the CE loss ($\delta_{\mathrm{ce}}$). Thus, to fit different adversarial data distributions, we introduce two BNs, each of which corresponds to one adversary type. Besides, we use the other BN for *normally transformed data*, i.e., $(\tau_1(x), \tau_2(x), x_\mathrm{h})$. Compared with existing work [26, 58, 59] that used 2 BNs (one for adversarial data and the other for benign data), our proposed ADVCL calls for triple BNs (TRIBN).

## C. Performance Summary under AutoAttack

In analogy to Figure 1 of the main paper, Figure A1 shows the performance comparison of ADVCL and baseline approaches with respect to (w.r.t) Auto-Attack Accuracy (AA) and Standard Accuracy (SA) both under SLF and AFF on CIFAR-10. We compare our proposed approaches with self-supervised pretraining baselines: AP-DPE [25], RoCL [27], ACL [26] and supervised adversarial training (AT) [10]. Upper-right indicates better performance w.r.t. standard accuracy and robust accuracy (under

**Algorithm A1** Pseudocode of Augmentation $\mathcal{T}$ in PyTorch.

```
transform = transforms.Compose([
    # random cropping
    transforms.RandomResizedCrop(size=32,
        scale=(0.2, 1.)),
    # random horizontal flip
    transforms.RandomHorizontalFlip(),
    # random color jittering
    transforms.RandomApply([
        transforms.ColorJitter(0.4, 0.4,
            0.4, 0.1)
    ], p=0.8),
    # random grayscale
    transforms.RandomGrayscale(p=0.2),
    transforms.ToTensor(),
])
```

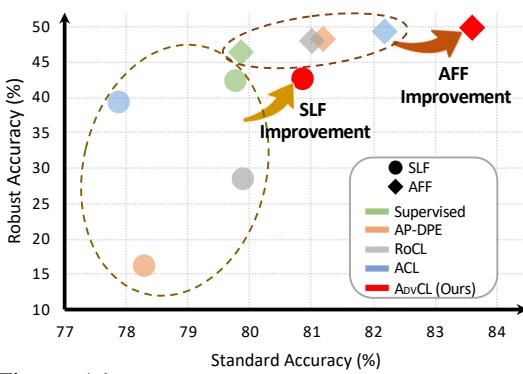

Figure A1: Performance of various robust pretrainig methods on CIFAR-10. Upper-right indicates better performance w.r.t. standard accuracy and robust accuracy (under Auto-Attack with $8/255$ $\ell_\infty$-norm perturbation strength).

Auto-Attack with $8/255$ $\ell_\infty$-norm perturbation strength). Colors represents pretraining approaches, and shapes represent finetuning settings. Circles (●) indicates *Standard Linear Finetuning* (SLF), and Diamonds (◆) indicates *Adversarial Full Finetuning* (AFF). As we can see, our proposed approach (ADVCL, red circle/diamond) has the best performance across both SLF and AFF settings and outperforms all baseline approaches significantly.

## D. Comparing SLF with ALF

Here we futher compare the performance gap of Standard Linear Finetuning(SLF) and Adversarial Linear Finetuning (ALF) with different pretraining approaches, by attacking the final model with various PGD attacks on CIFAR-10. The summarized results are in Figure A2. Different colors represent different pretraining approaches, and different line types represent different linear finetuning approaches. (**Solid line** for SLF and **dash line** for ALF). As we can see, ADVCL+SLF is already sufficient to outperform all baseline approaches under ALF. When the latter is applied to ADVCL, robustness is further improved by a small margin. This is different from baseline methods, where using ALF can boost the performance by a large margin. This phenomena justifies that the representations learned by ADVCL is more robust so that a standard finetuned linear classifier can already make the whole model robust, while the baseline approaches will need the linear classifier to be trained adversarially to obtain more satisfactory results. To the best of our knowledge, our approach is the only self-supervised pretraining approach that can outperform the supervised AT baseline under both SLF and ALF.

## E. Experiments on More Vision Datasets

We conducted more experiments on two in-domain settings: SVHN and TinyImageNet, and two cross-domain settings: SVHN→STL10 and TinyImageNet→STL10. We compare the performance of ADVCL with that of the other baseline methods in the Standard Linear Finetuning setup. The results are summarized in Table A3 and A4.

As we can see, our proposed ADVCL outperforms the other self-supervised and supervised adversarial training (AT) baselines in most cases, except for the in-domain TinyImageNet case on SA. However, the $0.2\%$ drop of SA corresponds to a more significant RA improvement of $3.4\%$. These results further justify the effectiveness of our proposal.

## F. Running Time Comparison

**Different Finetuning Method** Adversarial full finetuning (AFF) is much more computationally intensive than standard linear finetuning (SLF) per epoch. In Table A1, we list the detailed training time comparison between SLF and AFF. As we can see, AFF takes $24\times$ more training time than SLF.

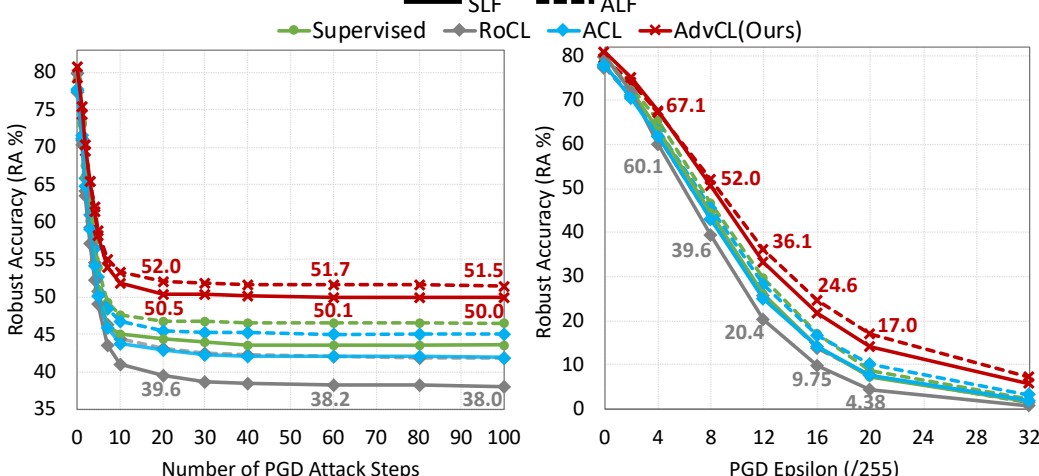

Figure A2: Robust accuracy (RA) of different pretraining approaches under various PGD attacks on CIFAR-10. SLF or ALF is applied to the pretrained model. Different colors represent different pretraining approaches, and different line types represent different linear finetuning approaches (solid line for SLF and dash line for ALF). Our proposed approach (ADVCL, red line) outperforms the baseline approaches in a non-trivial margin under all attack settings.

Table A1: Training time comparison for different finetuning approaches over a pretrained model.

| Finetuning Method | Running Time (per epoch × epochs) |
|---|---|
| Standard Linear (SLF) | 5.58s×25 |
| Adversarial Full (AFF) | 136.08s ×25 |

Table A2: Training time comparison for different pretraining approaches.

| Finetuning Method | Running Time (per epoch × epochs) |
|---|---|
| Supervised AT | 69.55s×200 |
| AP-DPE[25] | 6979.58s×150 |
| RoCL[27] | 123.15s×1000 |
| ACL[26] | 126.8s×1000 |
| ADVCL(ours) | 377.89s×1000 |

Table A3: Performance of ADVCL, compared with baselines, in terms of RA and SA on SVHN and Tiny-ImageNet, under Standard Linear Finetuning.

| Pretraining Method | SVHN | | TinyImageNet | |
|---|---|---|---|---|
| | RA(%) | SA(%) | RA(%) | SA(%) |
| Supervised | 41.03 | 91.13 | 16.03 | 42.73 |
| ACL[26] | 39.34 | 89.64 | 17.25 | 41.33 |
| ADVCL (ours) | 45.15 | 92.85 | 19.39 | 42.50 |

Table A4: Cross-dataset performance of ADVCL, compared with baselines, in terms of RA and SA, under Standard Linear Finetuning.

| Pretraining Method | SVHN→STL-10 | | TinyImgNet→STL-10 | |
|---|---|---|---|---|
| | RA(%) | SA(%) | RA(%) | SA(%) |
| Supervised | 13.81 | 38.97 | 24.31 | 60.93 |
| ACL[26] | 15.24 | 38.53 | 28.52 | 59.64 |
| ADVCL (ours) | 20.51 | 40.73 | 32.95 | 62.08 |

That is because AFF has to call for the min-max optimization (multiple inner-level maximization iterations needed per outer-level minimization step) to preserve model robustness.

**Different Pretraining Method**    We also demonstrate the training time costs of different pretraining methods in Table A2. We can make several observations:

1. Self-supervision-based pretraining methods take more time than the supervised AT method since self-supervised pretraining approaches typically require more epochs than fully supervised methods to converge.

2. In the self-supervision-based pretraining approaches, AP-DPE[25] takes the highest computation cost as it resorts to a complex min-max-based ensemble training recipe. Compared to the contrastive learning-based baselines (RoCL and ACL), ours (ADVCL) takes higher computation cost. This is because: (a) the contrastive loss of ADVCL takes more than two image views, and (b) ADVCL calls an additional pseudo supervision regularization. However, the pretraining procedure can often be conducted offline. Thus, the finetuning efficiency (via SLF) still makes ADVCL advantageous over the other baselines to preserve model robustness from pretraining to downstream tasks.

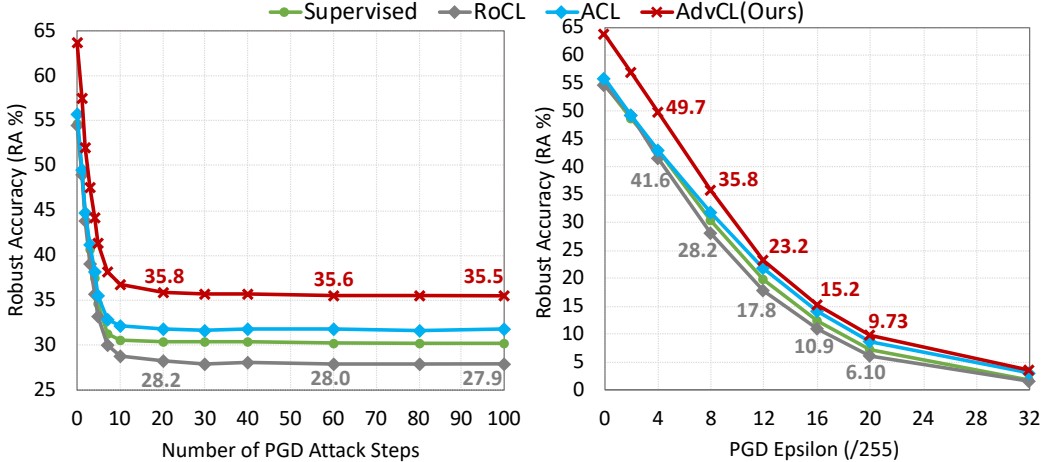

Figure A3: Robust accuracy (RA) of different pretraining approaches under various PGD attacks when transferring from CIFAR-10 to STL-10. SLF is applied to the pretrained model. Different colors represent different pretraining approaches. Our proposed approach (ADVCL, red) outperforms the baseline approaches in a non-trivial margin.

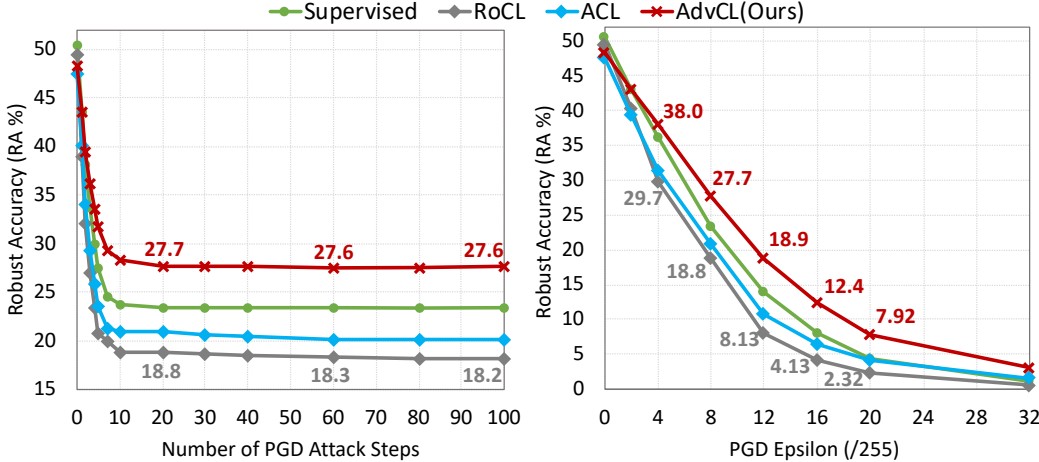

Figure A4: Robust accuracy (RA) of different pretraining approaches under various PGD attacks on CIFAR-100. SLF is applied to the pretrained model. Different colors represent different pretraining approaches. Our proposed approach (ADVCL, red) outperforms the baseline approaches in a non-trivial margin.

## G. Robust Transfer Learning

We also evaluate the model performance under various PGD attacks when transferring from CIFAR-10 to STL-10. Here SLF is applied to the pretrained model. We summarize the performance in Figure A3, where different colors represent different pretraining approaches. As we can see, ADVCL achieves the best performance under all attack settings and outperform baseline methods significantly.

## H. Ablation Studies on CIFAR-100

**Various attack strengths**   We also evaluate the models trained with different pretraining approaches under various PGD attacks on CIFAR-100, to further justify whether there exists the issue of obfuscated gradients on CIFAR-100. The summarized results are shown in Figure A4. As the results suggest, our ADVCL outperforms baseline approaches for most of the cases, especially when the attack becomes stronger with higher step or epsilon.