# OpenReview forum: "When does Contrastive Learning Preserve Adversarial Robustness from Pretraining to Finetuning?"
_NeurIPS.cc/2021/Conference — NeurIPS 2021 Poster_

### Official Review · Reviewer_5h6H · 2021-07-12

**Rating:** 5
**Confidence:** 4

**Summary:**

This work analyzed the fine-tuning phase of unsupervised contrastive learning for adversarial robustness. Furthermore, they proposed AdvCL which is more robust than other unsupervised adversarial contrastive learning methods. They demonstrated that the model trained with this method shows discriminative and interpretable representations, flat loss landscape, and better transferability to other tasks.

**Ethical Concerns:**

No ethical concerns.

**Limitations And Societal Impact:**

There is no limitation paragraph and exists a societal impact paragraph in Appendix. No negative societal impact. Several potential broader impact such as lightweight and efficient fine-tuning.

**Main Review:**

In this work, the objective is to address the issue of forgetting adversarial robustness while training the classifier with supervised learning after training feature encoder with unsupervised learning. They pointed out the different results between standard linear fine-tuning and adversarial full fine-tuning. However, the connection between the issue and the proposed method (AdvCL) is ambiguous. Only clusterfit seems to be related to the issue, but it is intuitive. Additionally, the method showed better performance than other methods when combined with standard linear fine-tuning, but showed similar performance when combined with adversarial full fine-tuning. They focused on the efficiency of standard linear fine-tuning and showed the higher performance of proposed method when considering efficiency, but it seems to be not efficient to calculate the cross-entropy loss perturbation and/or the contrastive loss perturbation respectively when using clusterfit.

The main adversarial contrastive loss (equation (7)) is suitable to make one adversarial perturbation using several augmentations, and HFC is the proper example. However, the connection issue still exists and only clusterfit seems proper. But, if this is included the proposed method (AdvCL + clusterfit), the method should show better performance than other method with clusterfit (RoCL + clutserfit, ACL + clusterfit, …). Additionally, also for the HFC, it should show better performance than other method with HFC (inserting HFC as augmentation or into positive set).

The discussions about the results are inadequate. Specifically, the discussions about the results of qualitative results and the ablation studies are necessary. It is required to discuss at least roughly what each result means and why.

Minor issues

(1) Because the obfuscated gradients issues can be checked using the performance of AutoAttack, there is no need to put the sanity check result on the main paper.

(2) In the method section, equation (4), it is confused that the input pair of similarity function of the denominator and the numerator. The inputs of the numerator may be $(\tau_1(x_1), \tau_2(x_1))$ and $(\tau_2(x_1), \tau_1(x_1))$, and the inputs of the denominator may be $(\tau_1(x_1), \tau_1(x_2))$, $(\tau_1(x_1), \tau_1(x_3))$,…,$(\tau_1(x_1), \tau_2(x_1))$, $(\tau_1(x_1), \tau_2(x_2))$ ,…, but cannot be sure.

(3) There is no the transferability results for cifar10 ↔ cifar100 as in the RoCL paper.


**Time Spent Reviewing:**

6 hours

---

> ### Author Response · Authors · 2021-08-08
> **Response to Reviewer 5h6H**
>
> We thank the reviewer very much for the insightful comments and suggestions.
>
> **Q1. [Connection between finetuning issues and AdvCL]**
> In the "self-supervised pretraining + supervised finetuning" paradigm, we observed that existing contrastive learning-based robust pretraining methods could not well preserve robustness under standard linear finetuning (SLF) (see Figure 1), which violates the generalization ability shown in standard contrastive learning. We think that there exist  two reasons for existing pretraining approaches to fail in SLF: **adversarial view mismatch** that lacks synergy between representation invariance and adversarial robustness, and **task mismatch** that makes feature representations learned from the pretraining task lacking robust generation to the supervised downstream task. Our proposed AdvCL is then designed to address the **above challenges**.
>
> In Table 4, we have made a careful ablation study on how the introduction of adversarial perturbations to the original example (rather than the transformed example used in the other baselines) and the use of high frequency components can help to gain adversarial robustness without losing standard accuracy. We also included the rationale behind these view selections in Lines 204 - 215, and 216-220. And In Table 5, we have made a detailed ablation study on the effect of ClusterFit. These experimental justifications show that both components (view selection and ClusterFit) in AdvCL are well-motivated and connected closely to preserve robustness from pretraining to finetuning.
>
> Response to the comment:
> >"They focused on the efficiency of standard linear fine-tuning and showed the higher performance of proposed method when considering efficiency, but it seems to be not efficient to calculate the cross-entropy loss perturbation and/or the contrastive loss perturbation respectively when using clusterfit."
>
> We would like to respectfully argue that our proposed AdvCL method is satisfied with the finetuning efficiency. At the pretraining stage, AdvCL requires *"to calculate the cross-entropy loss perturbation and/or the contrastive loss perturbation respectively when using clusterfit"*. However, the standard linear finetuning is built upon the fixed representation module of AdvCL and just needs supervised learning over the linear classification head, with no extra calculation. This is also recognized by Reviewer gD15:
> >*"Moving adversarial training toward less supervision (self-supervised, unsupervised) as well as making it easier to use (i.e. without full finetuning) are both of high interest for the machine learning community".*
>
> **Q2. [One adversarial perturbation vs. several augmentations]**
> First, as explained in Q1 (and justified in Table 4), to overcome the issue of **adversarial view mismatch**, it would be better to generate adversarial perturbation with respect to the original example rather than its transformation.
> Second, we conduct a new experiment that generates a universal adversarial perturbation applied to  both the original example $X$ and its high frequency component (HFC) $X_{HFC}$. We summarize our results on CIFAR-10 in the following table:
>
> <Table S1>. Performance (RA and SA) of AdvCL on CIFAR-10 under standard linear finetuning with different adversarial perturbation setups.
>
> | Method      | RA(%)  | SA(%)|
> | ----------- |:-----------:|:-----------:|
> | AdvCL  |50.45|80.85|
> | AdvCL + universal perturbation|48.03|79.37|
>
> As we can see, such a universal perturbation is not able to harmonize the adversarial robustness and generalizable contrastive representation. By contrast, $X_{HFC}$ itself might be proper without imposing perturbation. As we stated in Line 216-220, the reason is that: (1) learning over HFC of data is one of the main causes of achieving superior generalization ability, and (2) an adversary typically concentrates on HFC when manipulating an example to fool a model's decision.
>
> **Q3. [Comparison with  proper variants of baselines]**
> Thank you very much for suggesting multiple variants of baselines that we should consider. Following the reviewer’s suggestion, we consider a **new** set of experiments.
> First, we integrate the baseline methods (RoCL and ACL) with ClusterFit, yielding  RoCL + ClusterFit and ACL + ClusterFit. We then compare their performance with AdvCL to measure the significance of proper view selections adopted in AdvCL.
> Second, we expand the baseline methods using HFC as an additional viewpoint, leading to RoCL + HFC and ACL + HFC. We then compare their performance with AdvCL to demonstrate the importance of ClusterFit that we used. The following table summarizes RA and SA on CIFAR-10 using standard linear finetuning.
>
> <Table S2>. Performance (RA and SA) of AdvCL and proper variants of baselines on CIFAR-10 under standard linear finetuning.
>
> | Method      | RA(%)  | SA(%)|
> | ----------- |:-----------:|:-----------:|
> |RoCL|39.54|79.90|
> |ACL|42.87|77.88|
> |RoCL + HFC  |40.91|80.16|
> | ACL + HFC  |43.99|78.29|
> | ROCL + ClusterFit  |41.78|79.97|
> | ACL + ClusterFit  |44.52|78.12|
> | **AdvCL(ours)**  |**50.45**|**80.85**|
>
> As we can see, adding HFC or ClusterFit to existing baseline approaches could help improve their performance. This actually verifies the effectiveness of each proposed component in AdvCL. When comparing with these baseline variants, AdvCL still outperforms them, showing the necessity of both view-selection and pseudo-supervision in our AdvCL system. As shown in the results, our method still achieves the best performance. In the revision, we will include more experiments vs. the choice of model architectures.
>
>
>
>
> **Q4. [Enriched discussion on qualitative results and ablation studies]**
> Thanks for your comment. We will add more details on qualitative results (Sec. 5.2) and ablation studies (Sec. 5.3). Some key points and rationale are summarized below.
>
> In Sec. 5.2, we adopted 3 different visualizations: representation visualization, feature inversion map (FIM) visualization, and loss landscape visualization, to qualitatively show the advantage of our proposed AdvCL and provide some insights on why it is more robust.
>
> In Sec. 5.3, we designed 3 different experiment setups to show the advantage of each component of AdvCL. The first one shows the performance of AdvCL and baselines with different linear finetuning setups, the second one provides the performance using different contrastive views in pretraining, and the third one gives the  performance under different pseudo-label generating models.
>
> **Q5. [Minor Issues]**
> We thank the reviewer very much for the careful reading. We will address all of your minor comments in the revision.
>
>
> We wish that our response has addressed your concerns, and turns your assessment to the positive side. If you have any questions, please feel free to let us know during the rebuttal window. Thank you very much!

---

> > ### Author Response · Authors · 2021-08-21
> > **Look forward to your post-rebuttal feedback!**
> >
> > Dear Reviewer 5h6H,
> >
> > Thanks again for your insightful suggestions and comments. Since the deadline of discussion is approaching, we are happy to provide any additional clarification that you may need.
> >
> > In our previous response, we have carefully studied your comments and made detailed responses summarized below:
> > * Provided additional explanation on how AdvCL addresses finetuning challenges to preserve robustness.
> > * Conducted additional experiments to validate our choice of data augmentations.
> > * Conducted additional experiments to demonstrate the improvement of our proposal over a series of baseline variants that were suggested.
> > * Provided additional discussion on qualitative results and ablation studies.
> >
> > We hope that the provided new experiments and the additional explanation on robustness-accuracy trade-off have convinced you of the merits of our submission.
> >
> > Please do not hesitate to contact us if there's additional clarification or experiments we can offer. Thanks!
> >
> > Thank you for your time!
> >
> > Best, Authors

---

> > > ### Comment · Reviewer_5h6H · 2021-08-23
> > > **Question about the time complexity of the results**
> > >
> > > Thank you for reply.
> > >
> > > We agree that the standard linear finetuning results of the paper shows that the proposed method is better in maintaining the robustness, but the finetuning statge is short training compared to the pretraining stage. Thus, can you provide the training time results of Table 1 and Table 4 in the main paper? If possible, dividing them into pretraining and finetuning.

---

> > > > ### Author Response · Authors · 2021-08-24
> > > > **Additional results to address reviewer's new question**
> > > >
> > > > Thank you very much for your additional comments!
> > > >
> > > >
> > > > **First**, we would like to clarify that although the number of finetuning epochs could be much less than the number of contrastive-based pretraining epochs, adversarial full finetuning (AFF) is much more computationally intensive than standard linear finetuning (SLF) per epoch. In Table S3, we list the detailed training time comparison between SLF and AFF. As we can see, AFF takes **24$\times$** more training time than SLF. That is because AFF has to call for the min-max optimization (multiple inner-level maximization iterations needed per outer-level minimization step) to preserve model robustness. All experiments are performed on 4 NVIDIA TITAN Xp GPUs for a fair comparison.
> > > >
> > > >
> > > > **Table S3.** Training time comparison for different finetuning approaches over a pretrained model.
> > > >
> > > > |Finetuning Method | Time (per epoch $\times$ Epochs) |
> > > > | ----------- |:-----------:|
> > > > |SLF|5.58s $\times$ 25|
> > > > |AFF|136.08s $\times$ 25|
> > > >
> > > >
> > > > **Second**, we follow the reviewer's suggestion to demonstrate the training time costs of different pretraining methods in Table S4. We can make several observations:
> > > >
> > > > * Self-supervision-based pretraining methods take more time than the supervised AT method since self-supervised pretraining approaches typically require more epochs than fully supervised methods to converge.
> > > >
> > > > * In the self-supervision-based pretraining approaches, AP-DPE [1] takes the highest computation cost as it resorts to a complex min-max-based ensemble training recipe. Compared to the contrastive learning-based baselines (RoCL and ACL), ours (AdvCL) takes higher computation cost. This is because: (a) the contrastive loss of AdvCL takes more than two image views, and (b) AdvCL calls an additional pseudo supervision regularization. However, the pretraining procedure can often be conducted offline. Thus, the finetuning efficiency (via SLF) still makes AdvCL advantageous over the other baselines to preserve model robustness from pretraining to downstream tasks.
> > > >
> > > >
> > > > **Table S4.** Training time comparison for different pretraining approaches. (i.e. Table 1 in main paper)
> > > >
> > > > |Pretraining Method | Time (per epoch $\times$ Epochs) |
> > > > | ----------- |:-----------:|
> > > > |Supervised AT|69.55s $\times$ 200*|
> > > > |AP-DPE [1]|6979.68s $\times$ 150|
> > > > |RoCL [2]|123.15s $\times$ 1000|
> > > > |ACL [3]|126.82s $\times$ 1000|
> > > > |AdvCL (ours)|377.89s $\times$ 1000|
> > > >
> > > > \* For Supervised AT baseline it is not beneficial to train it for as many as 1000 epochs due to robustness overfitting. Therefore we follow the setting in [4] and only train it for 200 epochs. For the other baseline methods, the number of training epochs is set the same as their original experiment setting reported in their papers.
> > > >
> > > > **Third**, we follow the reviewer's suggestion to compare the computation time of contrastive learning-based pretraining under different view selection schemes. We list the detailed comparison in Table S5. As we can see, the use of more contrastive views will lead to a higher computation cost.
> > > >
> > > > **Table S5.** Training time comparisons under different contrastive view selections. (i.e. Table 4 in main paper)
> > > >
> > > > |Contrastive Views | Number of views | Time (per epoch $\times$ Epochs) |
> > > > | ----------- |:-----------:|:-----------:|
> > > > |$\tau_1(x)+\delta_1, \tau_2(x)$ |2|243.91s $\times$ 1000|
> > > > |$\tau_1(x)+\delta_1, \tau_2(x)+\delta_2$ |2 |260.33s $\times$ 1000|
> > > > |$\tau_1(x)+\delta_1, \tau_2(x)+\delta_2, \tau_1(x), \tau_2(x)$|4|407.50s $\times$ 1000|
> > > > |$x+\delta, \tau_1(x), \tau_2(x)$|3|314.65s $\times$ 1000|
> > > > |$x+\delta, \tau_1(x), \tau_2(x), x_l$|4|371.06s $\times$ 1000|
> > > > |$x+\delta, \tau_1(x), \tau_2(x), x_l, x_h$|5|445.39s $\times$ 1000|
> > > > |$x+\delta, \tau_1(x), \tau_2(x), x_h$ **(ours)**|4|377.89s $\times$ 1000|
> > > >
> > > > [1] Chen, T., Liu, S., Chang, S., Cheng, Y., Amini, L. and Wang, Z., 2020. Adversarial robustness: From self-supervised pre-training to fine-tuning. In CVPR 2020.
> > > >
> > > > [2] Kim, M., Tack, J. and Hwang, S.J., 2020. Adversarial self-supervised contrastive learning. In NeurIPS 2020.
> > > >
> > > > [3] Jiang, Z., Chen, T., Chen, T. and Wang, Z., 2020. Robust Pre-Training by Adversarial Contrastive Learning. In NeurIPS 2020.
> > > >
> > > > [4] Guo, M., Yang, Y., Xu, R., Liu, Z. and Lin, D., 2020. When nas meets robustness: In search of robust architectures against adversarial attacks. In CVPR 2020.

---

> > > > > ### Author Response · Authors · 2021-08-27
> > > > > **Looking forward to your response on the new experiment results**
> > > > >
> > > > > Dear Reviewer 5h6H,
> > > > >
> > > > > Thank you again for the insightful comments and suggestions!
> > > > >
> > > > > In our last response, we conducted additional experiments to demonstrate the training time cost of different pretraining and finetuning methods. For your convenience, we provide a summary below (see detailed results at the [previous response](https://openreview.net/forum?id=70kOIgjKhbA&noteId=WZPvmDSdREe)):
> > > > >
> > > > > * Conducted the new experiment in Table S3 to show that SLF is much more computationally light than AFF.
> > > > > * Conducted the new experiment in Table S4 to show the training time difference for different pretraining methods.
> > > > > * Conducted the new experiment in Table S5 to provide more insights into the connection between training time and different contrastive view configurations.
> > > > >
> > > > > Since there are only very few days left, we sincerely look forward to your follow-up response. If you have any new questions or experiment requests, please feel free to let us know. We are happy to provide additional explanations and evidence to illustrate the strength of our approach.
> > > > >
> > > > > Meanwhile, we would like to thank the reviewer again for the very helpful comments. By taking them into account, it would indeed make our paper clearer and stronger.
> > > > >
> > > > > Thank you again for your time!
> > > > >
> > > > > Best, Authors

---

### Official Review · Reviewer_gD15 · 2021-07-12

**Rating:** 7
**Confidence:** 4

**Summary:**

The submission investigates the adversarial robustness of self-supervised pretraining using contrastive learning with the NT-Xent loss [29]. The goal is to improve robustness of the pretraining step such that a linear classifier suffices to be fine-tuned for state-of-the-art performance, and to increase robustness across tasks. The proposed design, AdvCL, extends the loss formulation to contain four designed views (Eq. 9), as well as a supervision stimulus based on K-means clustering [50]. Experimental results show that the resulting system significantly increases adversarial robustness in the self-supervised setting. The paper also contributes a visualization of qualitative results as well as an ablation study of some of the key parameters.

**Limitations And Societal Impact:**

Yes, the authors have discussed the limitations of their work briefly in the conclusions, and the societal impact (which, in the eyes of this reviewer is not even applicable to this kind of work and amounts to guesswork) in the supplementary material.

**Main Review:**

The submission builds upon both adversarial training [10, etc.] as well as contrastive learning [29] for its loss and min-max training formulation, and extends these in the ways described above. It is not the first work to tackle the problem of self-supervised adversarial training (the authors cite [26, 27] as relevant related work). Overall originality is medium, since the proposed approach is a novel combination of existing elements -- but this is not a problem, since the resulting system is both well-designed and reasonably well motivated.
Moving adversarial training toward less supervision (self-supervised, unsupervised) as well as making it easier to use (i.e. without full finetuning) are both of high interest for the machine learning community. The paper does not offer any significant insights on adversarial training processes itself, but overall relevancy quite high.

The paper is generally very well written and easy to understand. Its sections are also well structured: introduction and related work lead to a clear formulation of the research questions, then the proposed system design, and finally a detailed evaluation.

The contributions themselves are well motivated and experimental results are overall convincing. However, I would have wished to see an ablation study on parameter \lambda; what happens when the supervision stimulus is not used at all, for example? Or how does the system perform without the proposed view selection? Separating these two contributions clearly in the experimental evaluation would have made the extent of each contribution somewhat clearer.

Overall this submission makes valuable contributions to advancing adversarial robustness in the laid out scenario and is of high enough quality to qualify for NeurIPS acceptance.
A few clarifications would make the submission even stronger:
- I thank the authors for providing footnote 1, which seems prudent. Does this footnote also imply that all presented experimental results were reproduced using either own implementations or code provided by the authors?
- The experimental results are almost all provided on CIFAR-10 or CIFAR-100, whereas experimental results on (additionally) ImageNet would increase confidence in the proposed method. I do see ImageNet results in Table 5. What was the reason for not including ImageNet experiments (or experiments on yet another large-scale, more realistic data set) in the main results? Is this connected to the scalability limitations described on line 377?
- What are the limitations described on line 377 in particular?

Minor notes:
- In lines 147 and 149, both \theta and f_theta are described as "feature encoder". I recommend describing the former as "feature encoder parameters".
- Both Equations 9 and 10 contain a minimization over \theta. I believe this is superfluous in Equation 9; otherwise it would be a double minimization. And the loss term itself should not contain the type of optimization.

There are some typos which could have been avoided by a judicious spell check:
- l. 42: 'fintuning' -> 'finetuning'
- Caption Figure 1: 'pretrainig' -> 'pretraining'
- l. 137: A citation seems to be missing.
- l. 141: 'pretrianing' -> 'pretraining'
- l. 188: 'mutli-view' -> 'multi-view'


**Time Spent Reviewing:**

3

---

> ### Author Response · Authors · 2021-08-08
> **Response to Reviewer gD15**
>
> We thank the reviewer very much for the positive comments and insightful suggestions.
>
> **Q1. [Ablation study on $\lambda$]**
> As we discussed in *"Sec. B. Implementation Details"* of the supplement, $\lambda$ is set to 0.2 in all our experiments. Such a parameter configuration is justified by a greedy search on the range of [0, 0.5]. We provide the missing ablation study in the following table:
>
> <Table S1>. Performance (RA and SA) of AdvCL on CIFAR10 under standard linear finetuning with different $\lambda$.
>
> | $\lambda$      | 0  | 0.05| 0.1| 0.2| 0.3| 0.5|
> | ----------- |:-----------:|:-----------:|:-----------:|:-----------:|:-----------:|:-----------:|
> | RA(%)  |48.89|49.34|49.96|**50.45**|50.23|50.08|
> | SA(%)  |77.73|79.68|80.03|**80.85**|80.39|79.98|
>
> **Q2. [Ablation studies on the absence of view selection and supervision stimulus ]**
> The reviewer suggests studying the effectiveness for each of the two components: view selection and pseudo-supervision. In fact, we have done these ablation studies in Table 4 and 5. More specifically, Table 4 reveals the importance of selected views, additive perturbation to the original example (4th row of Table 4), and high frequency components (last row of Table 4). Table 5 studied the necessity of supervision stimulus. The first row of Table 5 corresponds to pretraining without pseudo supervision (setting $\lambda=0$). Compared to such a baseline, the use of pseudo-supervision can boost RA by 1.6% and SA by 3.1%. We will add the above clarification in the revision.
>
>
> **Q3. [Baseline implementation]**
> We thank the reviewer for paying attention to the reproducibility. We are very cautious about our experiment setup to make sure the comparison is fair. For all baseline approaches, we either directly use their published model or train the model by their own published codes (if there is no published model in such settings), which should already be tuned by original authors to achieve the best performance.
>
> **Q4. [ImageNet Results]**
> The ImageNet in Table 5 was introduced to acquire a surrogate model for the pseudo-label generation, while our AdvCL pretraining has not been performed over ImageNet yet. Indeed, performing adversarial self-supervised pretraining on ImageNet is valuable. However, we are not able to finish such experiments due to our limited computing resources. To alleviate this concern, we thus conduct **new pretraining experiments** on the datasets TinyImageNet and SVHN.
> The results of using standard linear finetuning are summarized in the following tables.
>
> <Table S2>. Performance of AdvCL, compared with supervised and self-supervised baselines, in terms of RA and SA on SVHN and TinyImageNet under standard linear finetuning.
>
> | Method      | SVHN RA(%)  | SVHN SA(%)|TinyImageNet RA(%)|TinyImageNet SA(%)|
> | ----------- |:-----------:|:-----------:|:-----------:|:-----------:|
> | Supervised  |41.03|91.13|16.03|**42.73**|
> | RoCL        |37.75|91.02|15.97|42.68|
> | ACL         |39.34|89.64|17.25|41.33|
> | **AdvCL(ours)**|**45.15**|**92.85**|**19.39**|42.50|
>
> <Table S3>. Cross-dataset performance of AdvCL, compared with supervised and self-supervised baselines, in terms of RA and SA under standard linear finetuning.
>
> | Method      | SVHN->STL10 RA(%)  | SVHN->STL10 SA(%)|TinyImageNet->STL10 RA(%)|TinyImageNet->STL10 SA(%)|
> | ----------- |:-----------:|:-----------:|:-----------:|:-----------:|
> | Supervised  |13.81|38.97|24.31|60.93|
> | RoCL        |13.95|39.81|25.79|60.82|
> | ACL         |15.24|38.53|28.52|59.64|
> | **AdvCL(ours)**|**20.51**|**40.73**|**32.95**|**62.08**|
>
> As we can see,  our proposed AdvCL outperforms the other self-supervised and supervised adversarial training (AT) baselines in most cases, except for the in-domain TinyImageNet case on SA. However, the ~0.2% drop of SA corresponds to a more significant RA improvement of ~3.4%. These results further justify the effectiveness of our proposal.
>
> We remark that scalability to ImageNet also remains an open question in the other reproducible adversarial self-supervision baselines, and that is exactly the limitations we described in L377 in the main paper.
>
> Spurred by the reviewer’s question, we believe that an interesting future research direction is to validate if AdvCL (and the other baselines) can be integrated with a "fast" version of the adversarial training algorithm, which relies on a much cheaper attack generation method, known as 1-step PGD attack or FGSM attack [1]. We will add the above discussion into the Conclusion and explore it in the future. This is a very insightful comment. Thank you!
>
> [1] Wong, E., Rice, L. and Kolter, J.Z., 2019, September. Fast is better than free: Revisiting adversarial training. In ICLR 2019.
>
> **Q5. [Minor Issues]**
> We thank the reviewer for pointing out the notation issues. We will describe $\theta$ with "feature encoder parameters" and rewrite Eq 9. and Eq. 10 for ease of  understanding. We will also fix the other typos.

---

> > ### Comment · Reviewer_gD15 · 2021-08-31
> > **Thank you for your additional insights**
> >
> > I want to thank the authors for their additional insights provided, also in response to the other reviews. The final paper submission will be even stronger with these. I maintain that this is a good paper and support acceptance.

---

> > > ### Author Response · Authors · 2021-09-03
> > > **Thank you**
> > >
> > > Dear Reviewer gD15,
> > >
> > > Thank you very much for your support. We will surely revise our paper based on the responses to your comments.
> > >
> > > Thanks,

---

### Official Review · Reviewer_NaYZ · 2021-07-16

**Rating:** 7
**Confidence:** 4

**Summary:**

This paper proposes a unified adversarial contrastive learning framework. It proposes to improve the previous methods in the following ways: 1. Besides the previous augmentation methods, it proposes to utilize the high-frequency components as data augmentation to enhance the robustness. 2. It introduces the pseudo label for the pre-training task to mitigate the task difference between pre-training and fine-tuning for improving the transferability of the pre-training network. Many experiment result verifies the effectiveness of the proposed method. The author also offers ablation experiments to demonstrate the improvement brought by each component.

**Ethical Concerns:**

No ethical concerns.

**Limitations And Societal Impact:**

I don't see other limitations except for the points I mentioned in the main review.

No negative societal impact.

**Main Review:**

Originality: This paper employs the pseudo-class prediction task for pre-training and shows it can largely improve the transferability of the pre-training model. This is novel to me. I only saw the literature employing the pre-training task as regularization during fine-tuning.

Quality: The quality is overall good. Many experiments verify the effectiveness of the proposed method. Ablation studies also show the gain comes from each component. It would be better to see its performance on different backbones. There is only resnet18 here.

Clarity: The paper is overall well written and easy to follow.

Significance: The methods proposed in this paper significantly improve the transferability of the adversarial pre-trained model. Contributing to building a universal robust pre-training model.

Other: While the method requires a $\lambda$ to balance two loss terms, the value of $\lambda$ is not discussed.


**Time Spent Reviewing:**

3

---

> ### Author Response · Authors · 2021-08-08
> **Response to Reviewer NaYZ**
>
> We thank the reviewer very much for the positive comments and insightful suggestions.
>
>
> **Q1. [More neural network  architectures]**
> Following this suggestion, we conducted a **new experiment** to verify the effectiveness of our proposed approach on CIFAR10 using WideResNet-28-10. The achieved results are summarized in the following table:
>
> <Table S1>. Performance of AdvCL, compared with supervised AT and self-supervised baselines, in terms of RA and SA using WideResNet-28-10 on CIFAR-10, and standard linear finetuning.
>
> | Method      | RA(%)  | SA(%)|
> | ----------- |:-----------:|:-----------:|
> | Supervised  |46.26|85.95|
> | RoCL        |43.72|86.02|
> | ACL         |45.33|84.42|
> | **AdvCL(ours)**|**53.75**|**86.71**|
>
> As shown in the results, our method still achieves the best performance. In the revision, we will include more experiments vs. the choice of model architectures.
>
> **Q2. [Ablation study on $\lambda$]**
> As we discussed in *"Sec. B. Implementation Details"* of the supplement, $\lambda$ is set to 0.2 in all our experiments. Such a parameter configuration is justified by a greedy search on the range of [0, 0.5]. We provide the missing ablation study in the following table:
>
> <Table S2>. Performance (RA and SA) of AdvCL on CIFAR10 under standard linear finetuning with different $\lambda$.
>
> | $\lambda$      | 0  | 0.05| 0.1| 0.2| 0.3| 0.5|
> | ----------- |:-----------:|:-----------:|:-----------:|:-----------:|:-----------:|:-----------:|
> | RA(%)  |48.89|49.34|49.96|**50.45**|50.23|50.08|
> | SA(%)  |77.73|79.68|80.03|**80.85**|80.39|79.98|

---

> > ### Comment · Reviewer_NaYZ · 2021-08-12
> > **Question about the SLF result for WideResNet-28-10**
> >
> > Thanks for your reply.
> >
> > I noticed that the SLF performance of the supervised WideResNet-28-10 is much lower than the end-to-end trained supervised model. This is weird to me: Why tuning the linear classification layer on the same dataset could lead to such large performance dropping? I also found this is the same case for the results in the paper: the AFF results for the supervised model are always much higher than the SLF results. This is confusing. Since the backbone of the supervised has been fully trained with the labels, it should not lead to such a big difference between AFF and SLF. Could you please explain this? I think a possible reason could be the hyperparameters are not fully tuned for SLF. Linear evaluation sometimes involves different hyperparameters. For example, lr 30, wd 0 are employed for moco V2.

---

> > > ### Author Response · Authors · 2021-08-13
> > > **Further response**
> > >
> > > Thank you for your prompt reply!
> > >
> > > **[Q: The SLF performance of the supervised WideResNet-28-10 is much lower than the end-to-end trained supervised model? And SLF vs. AFF?]**
> > >
> > > (a) Let us first clarify the notations of pre-training methodologies used in Table S1, where all pre-trained models are finetuned using SLF as we stated in the previous response. Thus,  `Supervised` in Table S1 refers to the method of `Supervised AT pretraining + SLF`.
> > >
> > > (b) Based on our best understanding of the reviewer's comment, we assume that the following holds:
> > >
> > > (b1) The `SLF performance of the supervised WideResNet-28-10` is associated with the method of `Supervised AT pretraining + SLF`, namely, the  `Supervised` row of Table S1.
> > >
> > > (b2) The `end-to-end trained supervised model` refers to the model acquired  using  the conventional end-to-end supervised AT (without using  pretraining + finetuning).
> > >
> > > **We hope that our understanding of your comment in the above points (a) and (b) is correct. If not, please feel free to correct us, especially for (b2). Our response to your question is unfolded below.**
> > >
> > > **(R1)** First, we want to point out that the SLF performance of the `supervised WideResNet-28-10` (namely, the  `Supervised` row of Table S1) is comparable  to the `end-to-end trained supervised model` (using AT) reported in related publications, as shown  in **3rd row of Table 1 in [1]; 2nd row of Table 1 in [2]** (see reference details at the end of response): The `end-to-end trained supervised model` under WideResNet-28-10 (without using extra data) achieves 47.10% RA and 86.43% SA on CIFAR-10  in [1] (similarly found in [2]), which is quite close to the SLF performance of the  `Supervised` row of Table S1, with  46.26% RA and 85.95% SA, as shown in **Table S3**.
> > >
> > >
> > > **(R2)** Next,  we list the performance of the end-to-end Supervised AT model (reported in [1]),  the Supervised AT pretrained model, and our AdvCL pretrained model evaluated under both SLF and AFF settings, with different architectures on CIFAR-10. Note that as discussed in the *"Sec. B. Implementation Details"* of the supplement, we use the TRADES-type robust cross-entropy loss for AFF, following [3].
> > >
> > > **Table S3.** Performance of the end-to-end Supervised AT model (reported in [1]),  the Supervised AT pretrained model, and our AdvCL pretrained model evaluated under both SLF and AFF settings, with different architectures on CIFAR-10.
> > >
> > > | Method | Backbone | RA(%)  | SA(%)|
> > > | ----------- |:-----------:|:-----------:|:-----------:|
> > > |Supervised AT End-to-end [1] |ResNet-18|45.60|78.38|
> > > |Supervised AT + SLF  |ResNet-18|44.40|79.77|
> > > |Supervised AT + AFF |ResNet-18|49.89|79.86|
> > > |AdvCL(ours) + SLF|ResNet-18|50.45|80.85|
> > > |AdvCL(ours) + AFF|ResNet-18|52.77|83.62|
> > > |Supervised AT End-to-end [1] |WideResNet-28-10|47.10|86.43|
> > > |Supervised AT + SLF  |WideResNet-28-10|46.26|85.95|
> > > |Supervised AT + AFF |WideResNet-28-10|52.80|86.85|
> > > |AdvCL(ours) + SLF|WideResNet-28-10|53.75|86.71|
> > > |AdvCL(ours) + AFF|WideResNet-28-10|55.18|88.16|
> > >
> > > Based on the above table, we have the following observations:
> > >
> > > 1. `Supervised AT + SLF`    is quite comparable to  `Supervised AT End-to-end [1]` across models, comfirmed our previous response in **(R1)**
> > >
> > > 2. AFF outperforms SLF but **(2a)** the  performance of SLF drops mainly in  RA, and **(2b)** the gap in RA of SLF using our method (AdvCL) is **much smaller** than  `Supervised AT End-to-end [1]`, and the other self-supervised baselines (Table 1 in our paper).
> > >
> > >
> > > It is not surprising that SLF yields a worse RA than AFF, since SLF just learns a linear classification head using the standard training from scratch. By contrast, AFF applies the full (robustness-aware) adversarial training (AT) to the entire pretrained model using adversarial training loss. Thus, it will be more advantageous to RA improvement than SLF, as the former gives the entire classifier a  second chance to gain robustness from AT.
> > >
> > > Existing literature also observes the performance gap between SLF and AFF for different pretraining approaches. Comparing Line5 (corresponding to SLF in our case) and Line8 (corresponding to AFF) of [Table 3, 4], we can see that those pretrainig methods have an even larger performance gap between SLF and AFF (an average of ~**41.3% RA drop**). This clearly shows that *despite the existence of the gap between SLF and AFF, our method makes a significant and solid step to reduce it when using SLF.*
> > >
> > >
> > > We really appreciate the comment on the hyper-parameter choice. Following your suggestion,   we perform SLF using the same parameters (learning rate, momentum, weight decay, etc.) as Moco-v2 or SimCLR, and these parameters give us very similar results. This again justifies that preserving robustness is different from standard contrastive learning which only cares about SA.
> > >
> > >
> > > [1] Guo, M., Yang, Y., Xu, R., Liu, Z. and Lin, D., 2020. When nas meets robustness: In search of robust architectures against adversarial attacks. In CVPR 2020.
> > >
> > > [2] Zhang, H. and Wang, J., 2019. Defense against adversarial attacks using feature scattering-based adversarial training. In NeurIPS 2019.
> > >
> > > [3] Jiang, Z., Chen, T., Chen, T. and Wang, Z., 2020, October. Robust Pre-Training by Adversarial Contrastive Learning. In NeurIPS 2020.
> > >
> > >
> > > [4] Chen, T., Liu, S., Chang, S., Cheng, Y., Amini, L. and Wang, Z., 2020. Adversarial robustness: From self-supervised pre-training to fine-tuning. In CVPR 2020.

---

### Official Review · Reviewer_p6Bp · 2021-07-19

**Rating:** 6
**Confidence:** 3

**Summary:**

The focus of the paper is to make contrastive pre-training (e.g. SimCLR), which has seen great success in recent years, more robust to adversarial attacks. To this end, the authors propose AdvCL, which makes the following modifications to prior contrastive methods:
1) high-frequency components of images are shown to be important for robustness and are leveraged when building the views.
2) augmenting contrastive learning with "pseudo-supervision" obtained by clustering the features further improves performance.
The authors evaluate test accuracy on CIFAR10, CIFAR100, STL-10 under adversarial perturbations (AutoAttack and standard PGD) using both linear head fine-tuning and full fine-tuning and compare against recent SOTA methods. AdvCL seems to outperform the rest mostly across the board. Ablative studies are conducted.

**Ethical Concerns:**

No ethical concerns.

**Limitations And Societal Impact:**

Yes.

**Main Review:**

Overall, this paper checked all the boxes -- it was reasonably well-written, it had a solid experimental evaluation wherein the proposed method outperformed recent baselines, and ablations were conducted to shed more light on how AdvCL's inner workings. However, I have the following concerns:
1) Only 3 vision datasets were used, and in-domain results were only on 2 (CIFAR10 and CIFAR100). Furthermore, AdvCL loses to Supervised (control) on CIFAR100 standard accuracy (Table 1), yet the AdvCL entry is bolded -- why is this? This is also the case in table 2, CIFAR-100 --> STL-10. If this is indeed true, then improving adversarial robustness using the proposed method can sometimes hurt accuracy in the non-adversarial setting, which is concerning. I would like to see performance on SVHN, MNIST, and maybe a couple others.
2) Furthermore, the method is complex and begs the question -- are the gains reported really worth the complexity? It's quite feasible that many other alternatives of equal complexity and equal opportunity for hyper-parameter tuning can unlock the same gain. However, this may not be reason enough to reject the paper.
Point 1 is why I slightly lean against acceptance. If the authors elaborate on this and add a few more datasets, I would consider leaning in favor of acceptance.


**Time Spent Reviewing:**

1.5

---

> ### Author Response · Authors · 2021-08-08
> **Response to Reviewer p6Bp**
>
> We thank Reviewer p6Bp for the insightful comments and suggestions. We address each of your questions below.
>
> **Q1.1 [Robustness-accuracy Tradeoff]**
> It was an unintentional typo to highlight the AdvCL entry in the "SLF" row and the "(SA, CIFAR-100)" column of Table 1. The same typo appears in Table 2. We sincerely apologize for the confusion. We admit that the gain of adversarial robustness (in terms of RA) could yield a tradeoff with the loss of standard accuracy (SA). A similar observation holds for the self-supervised baseline ACL [1]; see the "CIFAR-100 --> STL-10" column of Table 2. In spite of a slight SA degradation, our proposed AdvCL yields a much more significant RA improvement than the other baselines. For the CIFAR100 in-domain setting, our approach drops 2.2% in SA but improves 4.3% in RA. For CIFAR100->STL10, our method drops in SA by only 0.4% while enhances RA by 6.7%. Furthermore, in the supervised adversarial training paradigm, it has been widely recognized that there should be a tradeoff between adversarial robustness and standard accuracy [2]. Our proposal in the "self-supervised pretraining + supervised finetuning" paradigm implies that such a tradeoff might not be completely mitigated in some cases (as the reviewer pointed out), but can be improved over supervised adversarial training, in terms of largely enhanced RA vs. slightly dropped SA.
>
> [1] Jiang, Z., Chen, T., Chen, T. and Wang, Z., 2020, October. Robust Pre-Training by Adversarial Contrastive Learning. In NeurIPS 2020.
>
> [2] Tsipras, D., Santurkar, S., Engstrom, L., Turner, A. and Madry, A., 2018, September. Robustness May Be at Odds with Accuracy. In ICLR 2018.
>
> **Q1.2 [More vision datasets]**
> Following the reviewer’s suggestion, we conducted **new experiments** on two in-domain settings: SVHN and TinyImageNet, and two cross-domain settings: SVHN->STL10 and TinyImageNet->STL10. We compare the performance of AdvCL with that of the other baseline methods in the Standard Linear Finetuning setup. The results are summarized in the following tables.
>
> <Table S1>. Performance of AdvCL, compared with supervised AT and self-supervised baselines, in terms of RA and SA on SVHN and TinyImageNet, under Standard Linear Finetuning.
>
> | Method      | SVHN RA(%)  | SVHN SA(%)|TinyImageNet RA(%)|TinyImageNet SA(%)|
> | ----------- |:-----------:|:-----------:|:-----------:|:-----------:|
> | Supervised AT  |41.03|91.13|16.03|**42.73**|
> | RoCL        |37.75|91.02|15.97|42.68|
> | ACL         |39.34|89.64|17.25|41.33|
> | **AdvCL(ours)**|**45.15**|**92.85**|**19.39**|42.50|
>
> <Table S2>. Cross-dataset performance of AdvCL, compared with supervised AT and self-supervised baselines, in terms of RA and SA, under Standard Linear Finetuning.
>
> | Method      | SVHN->STL10 RA(%)  | SVHN->STL10 SA(%)|TinyImageNet->STL10 RA(%)|TinyImageNet->STL10 SA(%)|
> | ----------- |:-----------:|:-----------:|:-----------:|:-----------:|
> | Supervised AT  |13.81|38.97|24.31|60.93|
> | RoCL        |13.95|39.81|25.79|60.82|
> | ACL         |15.24|38.53|28.52|59.64|
> | **AdvCL(ours)**|**20.51**|**40.73**|**32.95**|**62.08**|
>
>
> As we can see,  our proposed AdvCL outperforms the other self-supervised and supervised adversarial training (AT) baselines in most cases, except for the in-domain TinyImageNet case on SA. However, the ~0.2% drop of SA corresponds to a more significant RA improvement of ~3.4%. These  results further justify the effectiveness of our proposal.
>
>
> **Q2. [Gains vs.complexity]**
> This is a very good question. We believe that the complexity brought by our proposed pretraining method (AdvCL) is worthwhile considering its benefit to preserve adversarial robustness even using a lightweight standard linear finetuning on downstream tasks (see Figure 1 of our submission). A similar impact has also been recognized by Reviewer gD15:
> >"Moving adversarial training toward less supervision (self-supervised, unsupervised) as well as making it easier to use (i.e. without full finetuning) are both of high interest for the machine learning community."
>
> Besides the clarified value of our approach, the reviewer's question also motivates us to add a clearer discussion on the complexity of AdvCL vs. its resulting performance improvement.
> AdvCL consists of two components: view selection and pseudo-supervision stimulus. Each component is necessary and could boost the performance. For the view selection module, the multi-view contrastive loss can be efficiently implemented as the standard contrastive learning [3], together with the generation of adversarial examples as the supervised AT. Our ablation study in Table 4 shows the necessity of view selection, which boosts RA and SA by 9.1% and 7.4%, respectively. The pseudo-supervision stimulus is an effective and carefully designed mechanism to make learned features have class-discriminative power, without resorting to ground-truth labels. The ablation study in Table 5 shows the effectiveness and necessity of ClusterFit, where it gives us 1.6% and 3.1% improvement on RA and SA, respectively. Our code will be made publicly available for ease of reproduction.
>
> [3] Khosla, P., Teterwak, P., Wang, C., Sarna, A., Tian, Y., Isola, P., Maschinot, A., Liu, C. and Krishnan, D., 2020. Supervised Contrastive Learning. In NeurIPS 2020.
>
>
> We wish that our response has addressed your concerns, and turns your assessment to the positive side. If you have any questions, please feel free to let us know during the rebuttal window. Thank you very much!

---

> > ### Author Response · Authors · 2021-08-21
> > **Look forward to your post-rebuttal feedback!**
> >
> > Dear Reviewer p6Bp,
> >
> > Thanks again for your insightful suggestions and comments. Since the deadline of discussion is approaching, we are happy to provide any additional clarification that you may need.
> >
> > In the original review, you expressed the willingness to turn to the positive side if experiments on more vision datasets are offered to alleviate the concerns on the robustness-accuracy trade-off. We hope that the provided new experiments and the additional explanation on robustness-accuracy trade-off have convinced you of the merits of our submission.
> >
> > Please do not hesitate to contact us if there's additional clarification or experiments we can offer. Thanks!
> >
> > Thank you for your time!
> >
> > Best, Authors

---

> > > ### Comment · Reviewer_p6Bp · 2021-08-23
> > > **Changing my score from 5 -> 6**
> > >
> > > Thank you for providing extra experiments and thoroughly addressing each of my points. I will increase my score from 5 to 6.

---

> > > > ### Author Response · Authors · 2021-08-23
> > > > **Thank you!**
> > > >
> > > > Thank you very much for your response. We are glad to learn that your questions have been addressed.

---

### Author Response · Authors · 2021-08-08
**General Response: Contributions and New Experiments**

We thank all the reviewers very much for their insightful comments and constructive suggestions to strengthen our work. In addition to the response to specific reviewers, here we would like to highlight our contributions and the new experiments that we add in the rebuttal.

### Our Contributions
We are glad to find that reviewers generally appreciated and recognized our contributions:
* Making self-supervised adversarial training better to preserve adversarial robustness and easier to use under lightweight standard linear finetuning  [NaYZ, gD15];
* Having careful design and reasonable motivation of the proposed architecture [NaYZ, gD15];
* Having solid experiment results showing that our method outperforms baselines significantly [p6Bp, NaYZ, gD15];
* Having extensive ablation studies to demonstrate the usefulness of proposed model components [p6BP, NaYZ];
* Having paper well-written and easy to follow [p6Bp, NaYZ, gD15].

### New Experiments
In this rebuttal, we have added more supporting experiments following reviewers’ suggestions. We summarize all the experiments that we added below:
* More vision datasets [p6Bp, gD15].
* More backbone architectures [NaYZ].
* Ablation studies on $\lambda$ [NaYZ, gD15].
* Universal adversarial perturbation applied to multiple augmentations [5h6H].
* Comparison with  variants of baselines  [5h6H].

---

### Author Response · Authors · 2021-09-02
**Thank you! And a summary of our rebuttal and discussion**

We sincerely appreciate all reviewers’ and ACs’ efforts and time in reviewing our paper and pushing the rolling discussion. We truly thank you all for the insightful and constructive comments and suggestions, which helped us to further improve our paper and make it stronger. We truly appreciate the positive 7-7-6 evaluation from reviewers gD15, NaYZ, and p6Bp. We are also thankful for the insightful suggestions from reviewer 5h6H.

Here is a summary of our updates:
* **[Additional experiments to further justify the effectiveness of our approach]** As suggested by reviewers, we conduct extra experiments on more vision datasets, more backbone architectures, ablation studies on λ parameter, universal adversarial perturbation applied to multiple augmentations, comparison with variants of baselines, and comparisons on time complexity for both pretraining and finetuning. **@reviewer 5h6H, we also provided additional experiment results and analysis on your latest question on computation cost. Please feel free to check.**

* **[Manuscript revision]** We owe many thanks to reviewer gD15’s extremely helpful writing suggestions. All improved manuscript parts, together with other constructive discussions with all reviewers, will be delivered in our final version.

We really thank all reviewers’ and ACs’ time and efforts again.

Best wishes,

Authors

---

### Decision · Program_Chairs · 2021-09-27

**Decision:**

Accept (Poster)

**Comment:**

With the increasing interest in both adversarial robustness and contrastive self-supervised learning, the paper studies a highly relevant problem at their intersection: how contrastive learning (CL) can be made to learn representations so that robustness is preserved when fine-tuning a linear classification head on top of these representations.

This indeed seems to be a setting that hasn't been explored before in the literature; however, to my knowledge such a setting in which we use standard linear fine-tuning is not commonly used (previous works such as SimCLR only use it to prove a point about the power of the learned representations from CL alone). Thus, I am willing to grant it here as well, as this provides us more insight into the robustness of the CL learned representations, and although one reviewer pointed out concerns about the higher runtime of the proposed procedure, this point may not be so important in this context. Also, as one reviewer pointed out, this may help encourage more light-weight fine-tuning if indeed the robustness can be transferred across tasks making adversarial robustness easier to use.

Moreover, the paper also provides a thorough experimental evaluation not only on the standard linear fine-tuning but also on the adversarially robust fine-tuning, shows that the proposed method (which appears to be an extension of ACL), is superior to previous methods with ablation studies.